# Ceg1 depletion reveals mechanisms governing degradation of non-capped RNAs in *Saccharomyces cerevisiae*

Onofrio Zanin[1,2], Matthew Eastham[3], Kinga Winczura[3], Mark Ashe[3], Rocio T. Martinez-Nunez [2],
Daniel Hebenstreit[4] & Pawel Grzechnik [3✉]

Most functional eukaryotic mRNAs contain a 5′ 7-methylguanosine ($m^7G$) cap. Although capping is essential for many biological processes including mRNA processing, export and translation, the fate of uncapped transcripts has not been studied extensively. Here, we employed fast nuclear depletion of the capping enzymes in *Saccharomyces cerevisiae* to uncover the turnover of the transcripts that failed to be capped. We show that although the degradation of cap-deficient mRNA is dominant, the levels of hundreds of non-capped mRNAs increase upon depletion of the capping enzymes. Overall, the abundance of non-capped mRNAs is inversely correlated to the expression levels, altogether resembling the effects observed in cells lacking the cytoplasmic 5′—3′ exonuclease Xrn1 and indicating differential degradation fates of non-capped mRNAs. The inactivation of the nuclear 5′—3′ exonuclease Rat1 does not rescue the non-capped mRNA levels indicating that Rat1 is not involved in their degradation and consequently, the lack of the capping does not affect the distribution of RNA Polymerase II on the chromatin. Our data indicate that the cap presence is essential to initiate the Xrn1-dependent degradation of mRNAs underpinning the role of 5′ cap in the Xrn1-dependent buffering of the cellular mRNA levels.

[1] School of Biosciences, University of Birmingham, Edgbaston, Birmingham B15 2TT, UK. [2] School of Immunology & Microbial Sciences, King's College London, Guy's Campus, London SE1 9RT, UK. [3] Division of Molecular and Cellular Function, School of Biological Sciences, University of Manchester, Oxford Road, Manchester M13 9PT, UK. [4] School of Life Sciences, University of Warwick, Coventry CV4 7AL, UK. ✉email: pawel.grzechnik@manchester.ac.uk

The N7-methylated guanosine (m[7]G cap) linked to the first nucleotide of the RNA molecule, is a hallmark structure of eukaryotic transcripts generated by the RNA Polymerase II (Pol II). The cap is involved in various steps of mRNA turnover, processing, transport and translation and therefore, essential for cell viability[1, 2]. Many of these biological functions are related to the complexes interacting with the m[7]G caps: the nuclear cap-binding complex (CBC) and the cytoplasmic eukaryotic initiation factor 4E (eIF4E)[3–5]. The biosynthesis of the m[7]G cap requires three enzymes: the RNA triphosphatase (TPase) which removes the γ-phosphate from the RNA 5′ end; the RNA guanylyl-transferase (GTase) which transfers a GMP group to the diphosphate 5′ end; and the guanine-N7 methyltransferase adding a methyl group to the N7 amine of the guanine cap[6]. This synthesis pathway of the m[7]G cap is conserved in eukaryotes, however, the enzymes involved vary between the organisms[7]. In humans and other higher eukaryotes, a bifunctional protein RNGTT acts as RNA 5′-triphosphatase and guanylyltransferase while the methylation is mediated by a separate protein RNMT[8]. In *Saccharomyces cerevisiae,* three proteins are responsible for RNA capping: the 5′-triphosphatase Cet1 and the guanylyltransferase Ceg1 work together as a stable heterodimer or heterotetramer, while the methyltransferase Abd1 acts independently[9–11]. The capping enzymes are recruited to the Pol II at the early stage of transcription. In *S. cerevisiae*, Cet1 and Ceg1 bind the serine 5 phosphorylated C-terminal domain (CTD) of Pol II at the transcription start site (TSS)[9,12]. When the nascent RNA reaches the length of ~17 nt, the Ceg1-Cet1 complex docks to the Pol II surface close to the RNA exit tunnel[11] and adds the terminal guanosine as soon as the first 25–30 nt of the nascent transcript extrudes from the transcribing complex[13]. Changes in the phosphorylation status of the Pol II CTD promoting transcription elongation result in the rapid dissociation of Ceg1-Cet1[11]. This and the increased phosphorylation of CTD Ser2 recruit Abd1 about 110 nt downstream of the TSS[12] to complete the formation of the m[7]G cap.

If the methylation of the added 5′ guanosine residue fails, the defective transcripts are detected by Npl3-mediated surveillance mechanism that triggers decapping by Rai1 and subsequent RNA degradation by its co-factor, the 5′−3′ exonuclease Rat1[14,15]. The binding of CBC to the cap, which indicates correct and completed capping, disrupts Npl3-Rai1 and prevents degradation[15]. Nascent transcripts lacking the 5′ cap may be co-transcriptionally degraded. Such a process occurs at the 3′ ends of genes where cleavage over the poly(A) signal in the nascent RNA creates an unprotected 5′ monophosphate end which serves as an entry point for Rat1[16–18]. If the degradation rate exceeds the RNA synthesis rate, Rat1 reaches the transcribing complex and Pol II is displaced from DNA by so-called 'torpedo' termination[16–19]. A similar mechanism has been suggested to occur upstream of the polyA site when capping fails at the 5′ end of the gene. Analysis of the *FMP27* gene in *S. cerevisiae ceg1-63* mutant revealed that the transcription of uncapped pre-mRNA is prematurely terminated by Rat1 within the gene body which reduced the level of *FMP27* mRNA[20]. Analysis of the temperature-sensitive mutants showed that the inactivation of Ceg1 also affected the accumulation of heat-shock induced *SSA1* and *SSA4* mRNAs. Their levels were restored by the deletion of the predominantly cytoplasmic 5′−3′ exonuclease Xrn1[1]. Xrn1 mediates global mRNA degradation and acts with the decapping factors as a sensor of the cellular mRNA levels that controls and regulates the homoeostatic mRNA abundance[21,22].

Here, we investigated the consequences of the capping absence on RNA accumulation and transcription. We employed rapid nuclear depletion of Ceg1 to investigate the global turnover of non-capped mRNAs. We found that the lack of the capping resulted in the differential expression of cellular RNAs: the mRNA levels of highly expressed genes decreased while the mRNAs of genes expressed at lower levels increased indicating differential specificity of degradation pathways of highly and lowly expressed genes. The deletion of *XRN1* resulted in a similar redistribution in the mRNA abundance which was not synergistically affected by the nuclear depletion of Ceg1. Our data indicate that m[7]G cap and Xrn1 act together in the control of the mRNA 5′−3′ degradation. In particular, the presence of the cap is critical for the formation of the correct 5′ end substrate for efficient Xrn1-dependent degradation. The non-capped mRNA levels were not rescued by the inactivation of the nuclear 5′−3′ exonuclease Rat 1 and the disruption of the 5′ end capping had a minor effect on the distribution of Pol II on the chromatin. Overall, our data indicate that non-capped mRNAs, in contrast to defectively capped mRNAs, are not specifically marked for degradation and that m[7]G caps play important roles in the Xrn1-dependent degradation buffering cellular mRNA levels.

## Results

**Differential expression of non-capped mRNAs**. The impact of defective or complete lack of capping on RNA abundance has never been studied genome-wide before. In our experimental approach, we employed the anchor-away (AA) system[23] to deplete the capping enzymes from the nucleus and rapidly affect the cap formation. FRB tag fused with either Cet1, Ceg1 or Abd1 (*cet1-AA*, *ceg1-AA* and *abd1-AA* strains) in the presence of rapamycin heterodimerized with FKBP12 attached to the ribosomal protein Rpl13a, which was exported to the cytoplasm thus removing the capping enzymes from the nucleus. Since rapamycin affects the TOR signalling pathway[24] the strain used for this system carries the *tor1-1* mutation, which permits normal growth on rapamycin[23]. The nuclear depletions of Cet1 and Ceg1 were lethal (Fig. 1a) indicating the efficacy of the method as cap formation is essential for cell viability[25,26]. The inactivation of Abd1 activity has been reported to be non-viable[27], however, the nuclear depletion of Abd1 had a mediocre impact on cell growth (Supplementary Fig. 1a). This indicates that Abd1 may still perform its functions after the relocation to the cytoplasm. The nuclear depletion of either Cbp20 or Cbp80 was viable, consistent with the fact that the CBC is not essential in *S. cerevisiae* (Fig. 1a). In our functional experiments, the same yeast culture was split into a medium containing either vehicle DMSO or rapamycin which represented the WT (DMSO) or the nuclear depletion (Rap) condition (Fig. 1b). First, we tested the efficiency of Ceg1 and CBC subunit depletions. We confirmed by immunofluorescence visualisation of GFP-fused proteins that proteins were depleted from the nucleus after 45 min of rapamycin treatment (Fig. 1c, d). Next, we confirmed that RNA capping was indeed affected by the nuclear depletion of Ceg1. We co-immunoprecipitated RNA isolated from the *ceg1-AA* strain growing for 45 min on either DMSO or Rap with the anti-m[7]G cap antibody (H-20) and tested mRNAs levels using qPCR (Fig. 1e and Supplementary Fig. 1b). The amount of capped mRNAs (normalised to human mRNA *GAPDH* spike) was reduced up to 80% in *ceg1-AA* (growing on Rap) when compared to WT (control cells grown on DMSO). The nuclear depletion of either Ceg1 or Cet1 (*ceg1-AA* or *cet1-AA*) resulted in a similar decrease in the *ADH1* mRNA (which has been used in many studies as a model yeast mRNA) (Fig. 1f and Supplementary Fig. 1c); therefore, in the subsequent analyses, we focused mainly on the effects of Ceg1 nuclear removal. We also confirmed that the nuclear depletion of CBC did not affect *ADH1* mRNA and so the effect we observed during the depletion of capping machinery was solely caused by the lack of Ceg1/Cet1 function (Fig. 1f and

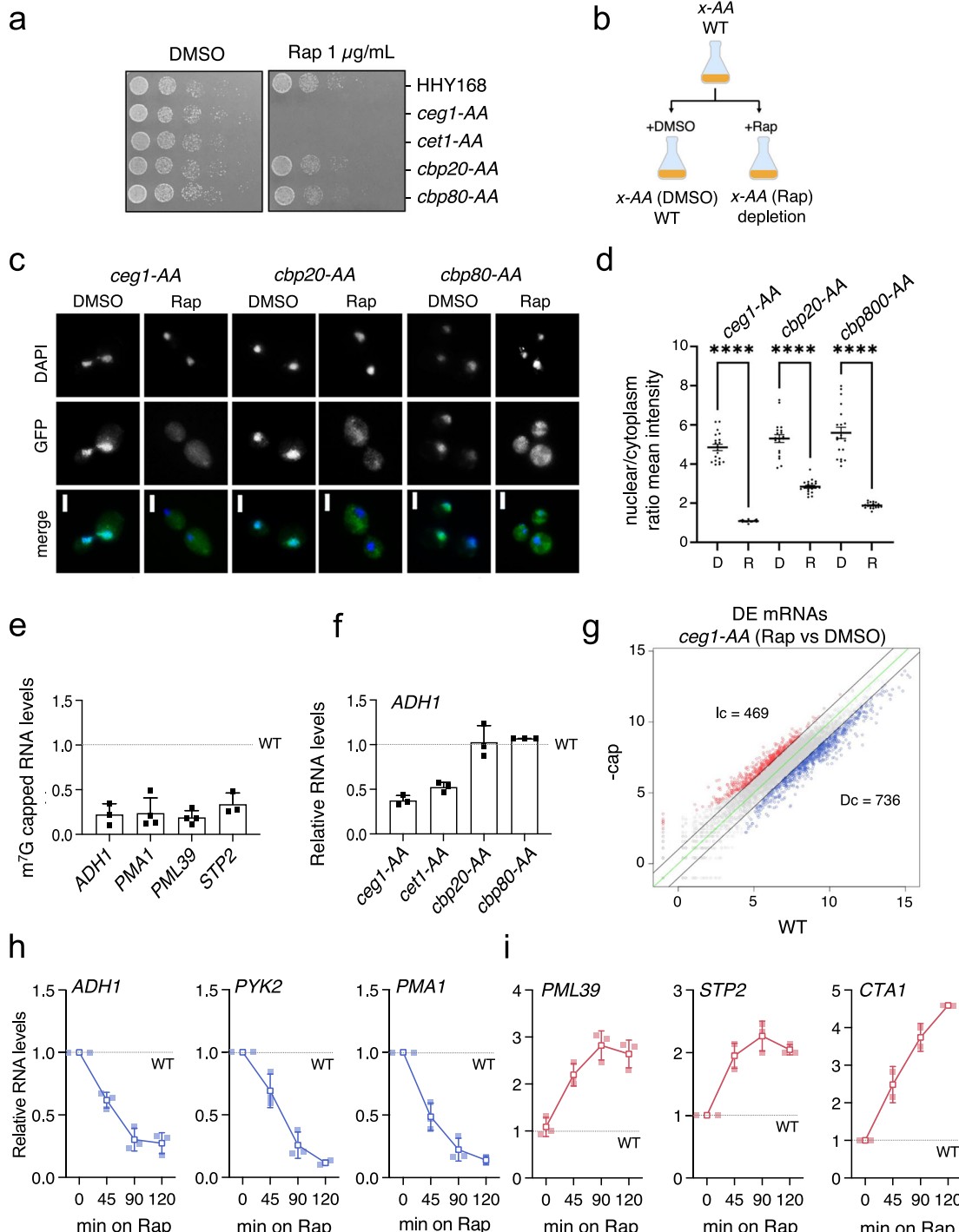

Supplementary Fig. 1c). Nuclear depletion of Abd1 did not change *ADH1* levels (Supplementary Fig. 1d).

Next, we nucleodepleted Ceg1 for 45 min and performed a global analysis of mRNA levels. Our RNA-seq analysis identified 1205 mRNAs affected by the nuclear depletion of Ceg1 (displaying log2 fold change <−1 and >1) (Fig. 1g). 736 (61%) of these mRNAs decreased while the levels of 469 genes (39%) increased when compared to the control (WT). Next, we tested how the mRNA levels changed over time during the nuclear depletion of Ceg1. We selected *ADH1*, *PYK2* and *PMA1* genes whose levels decreased upon Ceg1 depletion in the RNA-seq data. RT-qPCR analysis revealed that their mRNAs decreased even further after 90 min and stabilised after 120 min (Fig. 1h). We

also tested *PML39*, *STP2* and *CTA1* whose levels increased upon Ceg1 depletion (Fig. 1i). *PML39* and *STP2* mRNAs peaked after 90 min, while the accumulation of *CTA1*, displayed a constant increase at all time points. The increased and decreased mRNA levels were confirmed by RT-qPCR relative to exogenous spike-in control (Supplementary Fig. 1e). Yeast RNA was mixed 1:1 with human RNA and subjected to qPCR analysis where mRNA levels were normalised to 28S rRNA and reverse transcription reaction efficiency was corrected to human *GAPDH*.

**Accumulation of non-capped mRNAs depends on individual expression levels.** To understand the differential expression observed upon depletion of the capping machinery, we analysed

**Fig. 1 Nuclear depletion of capping enzymes results in differential accumulation of mRNAs. a** Spot test showing the growth of the AA strains. Serial dilutions of each strain were spotted on YPD media containing DMSO or rapamycin (1 μg/ml). **b** The outline of the experimental approach using AA strains. Cell cultures at the exponential phase were split and supplemented with DMSO (WT; control sample) or rapamycin (Rap; nuclear depletion sample). **c** Fluorescent microscopy localisation of GFP-AA tagged strains after 45 min incubation on DMSO or Rap. DAPI was used to visualise nuclei. White scale bars represent 2 μM. **d** Quantification of mean signals intensity of the cytoplasm and nucleus in GFP-AA strains growing on DMSO (D) or rapamycin (R). Statistical significance was determined through one way ANOVA. ** denotes $p < 0.01$, **** denotes $p < 0.0001$. **e** Quantification of the capped RNA by RT-qPCR after 45 min of rapamycin treatment compared to control (DMSO; WT). The RNA isolated from yeast was spiked-in with capped human RNA and immunoprecipitated by the anti-m7G cap antibody (clone H20). The RNA levels were normalised to human *GAPDH* mRNA. The error bars show standard deviation of three independent experiments. **f** RT-qPCR analysis of the *ADH1* mRNA levels in the AA-tagged strains after 45 min of rapamycin treatment compared to the control (isogenic strains on DMSO) set to 1 (dotted line). The error bars show standard deviation of two (*cbp20-AA* and *cbp80-AA*) or three independent experiments **g** RNA-seq analysis showing differential expression of mRNA species in *ceg1-AA* (Rap) strain (-cap) after 45 min of rapamycin treatment compared to the control condition (WT). Increased (Ic) RNA species with log2 fold change >1 are labelled in red. Decreased (Dc) RNA species showing log2 fold change <-1 are in blue. The average values with $p_{adj} < 0.05$ of $n = 2$ experiments are shown, the zero-change line is in green and the ±1 log2 fold change threshold is indicated with black lines. The changes in levels of decreased (**h**) and increased (**i**) mRNAs over time during nuclear depletion of Ceg1. RT-qPCR analysis of RNA levels for different genes measured at 0, 45, 90 and 120 min of rapamycin treatment relative to control (WT) set to 1 (dotted line). The error bars show standard deviation of three independent experiments.

the features of affected mRNAs. First, we tested the basal expression (levels in WT cells) of the differentially expressed mRNAs in Ceg1-depleted cells. We found that increased mRNAs were generally lower expressed (median transcript per million TPM log2 = 2.7) than the decreased mRNAs (median log2 TPM = 6.2) (Fig. 2a). Next, we split all 1205 protein-coding genes differentially expressed in *ceg1-AA* into three bins according to their expression levels in WT cells: the top 25% were classified as highly expressed (High), the bottom 25% as lowly expressed (Low) and the remaining 50% as medially expressed (Mid). Most of the lowly expressed genes increased their mRNA levels (291 out of 301), while most of the highly expressed mRNAs decreased (300 out of 301) after the Ceg1 depletion (Fig. 2b, c). Mid-range expressed mRNAs were the least affected by the depletion; however, they displayed a tendency to decrease in Ceg1-depleted cells (426 decreased versus 176 increased), correlating with increasing expression of individual genes. This was also evident at the single gene level as shown for representative *ZRT1*, *STB4* and *ADH1* (Fig. 2d). Next, we employed available datasets estimating transcription rates (TR) and Pol II density in *S. cerevisiae*[28] and applied them to the sets of differentially expressed mRNAs identified in the *ceg1-AA*. These analyses revealed that mRNAs which increased upon Ceg1 have lower transcription rates and Pol II density (Fig. 2e and Supplementary Fig. 2a, b) than genes whose mRNAs decreased in Ceg1-depleted cells. Consistently with the fact that shorter genes are generally associated with higher expression, the mRNAs accumulating in *ceg1-AA* were also longer than those from the decreased set (average gene length 1430 vs 1113 nt) (Supplementary Fig. 2c) additionally confirming the relation between the expression strength and stability of non-capped RNAs. The determination of the half-lives of affected mRNAs would additionally characterise these transcripts. However, available datasets estimating mRNA half-lives in yeast significantly vary depending on the method used in the studies (Supplementary Fig. 2d)[29,30], therefore such analysis was not conclusive.

We also investigated Pol II-transcribed non-coding RNAs (ncRNAs), called cryptic unstable transcripts (CUTs) and stable unannotated transcripts (SUTs), which in normal conditions are expressed at a very low level[31, 32]. Most CUTs are detectable only upon the inactivation of the 3′−5′ exonuclease Rrp6 subunit of the nuclear exosome[33] and for this reason, only a handful of CUTs were detected in our RNA-seq analysis of the *ceg1-AA* strain. Thus, we focused our investigation on the well-studied ncRNA *CUT542* (*NEL025c*). The nuclear depletion of Ceg1 increased the levels of *NEL025c* (Fig. 2f) up to ten times after 120 min of rapamycin treatment, consistent with the fact that

lowly expressed Pol II transcripts accumulate when capping enzymes are not functional. A similar effect was observed for *SUT238* (Fig. 2f).

Next, we tested whether the alternation in the RNA synthesis could rescue the levels of *ADH1* mRNA and *NEL025c* ncRNA in *ceg1-AA*. We affected Pol II elongation rate[34,35], by treating cells with 6-azauracil (6-AU) for 30 min, which causes depletion of intracellular nucleotide pools[36]. The treatment was followed by Ceg1 nuclear depletion and analysis of the *ADH1* and *NEL025c* levels relative to the WT (Fig. 2g). Slower RNA synthesis stabilised *ADH1* mRNA however, did not affect the over-expression of *NEL025c* which suggests that the low expression of this gene reached the threshold where further decrease had minimal impact on the accumulation of non-capped RNA.

Highly expressed genes are characterised not only by the strength of the promoter but also their sequences which determine the efficiency of translation or degradation pathways. The set of genes that decreased upon Ceg1 nuclear depletion displayed a higher GC content than the increased set (Fig. 3a) as the high GC content is associated with high expression[37]. Moreover, we found that the increased and decreased mRNA groups also display a different codon bias in their sequences (Fig. 3b). This is consistent with the fact that codon usage for highly expressed genes has a high correlation with tRNA abundance and thus differs from the overall characteristics of the whole yeast genome[38] (Supplementary Fig. 3). Both features may contribute to the differential fate of the transcripts, including RNA storage and selection of the degradation pathways[39–41]. For example, decapping enzymes and Xrn1 preferentially bind transcripts containing a low number of optimal codons while the deadenylation enzymes and the 3′−5′ exonucleolytic complex exosome bind more strongly to codon optimal transcripts[40]. Finally, utilising Multiple Em for Motif Elicitation (MEME, https://meme-suite.org/meme/) analysis identified the sequence motif (A/G)GAAAA strongly enriched and uniformly distributed in the upregulated mRNAs (Fig. 3c). The motif resembles the binding sequence of the human NFAT (nuclear factor of activated T-cells) factor involved in T-cells immune response[42], displaying functional similarity with the transcription factor Crz1 in *S. cerevisiae*[43]. However, the function of the identified sequence in the context of RNA stability remains unknown.

**Minor contribution of Rat1 in the degradation of non-capped Pol II transcripts**. Next, we tested how non-capped mRNAs are degraded. Previous reports showed that mutation of the exonuclease Rat1 can delay degradation of *FMP27* and *GAL1* mRNAs in capping-deficient cells[20] and that the Rai1/Rat1 complex is

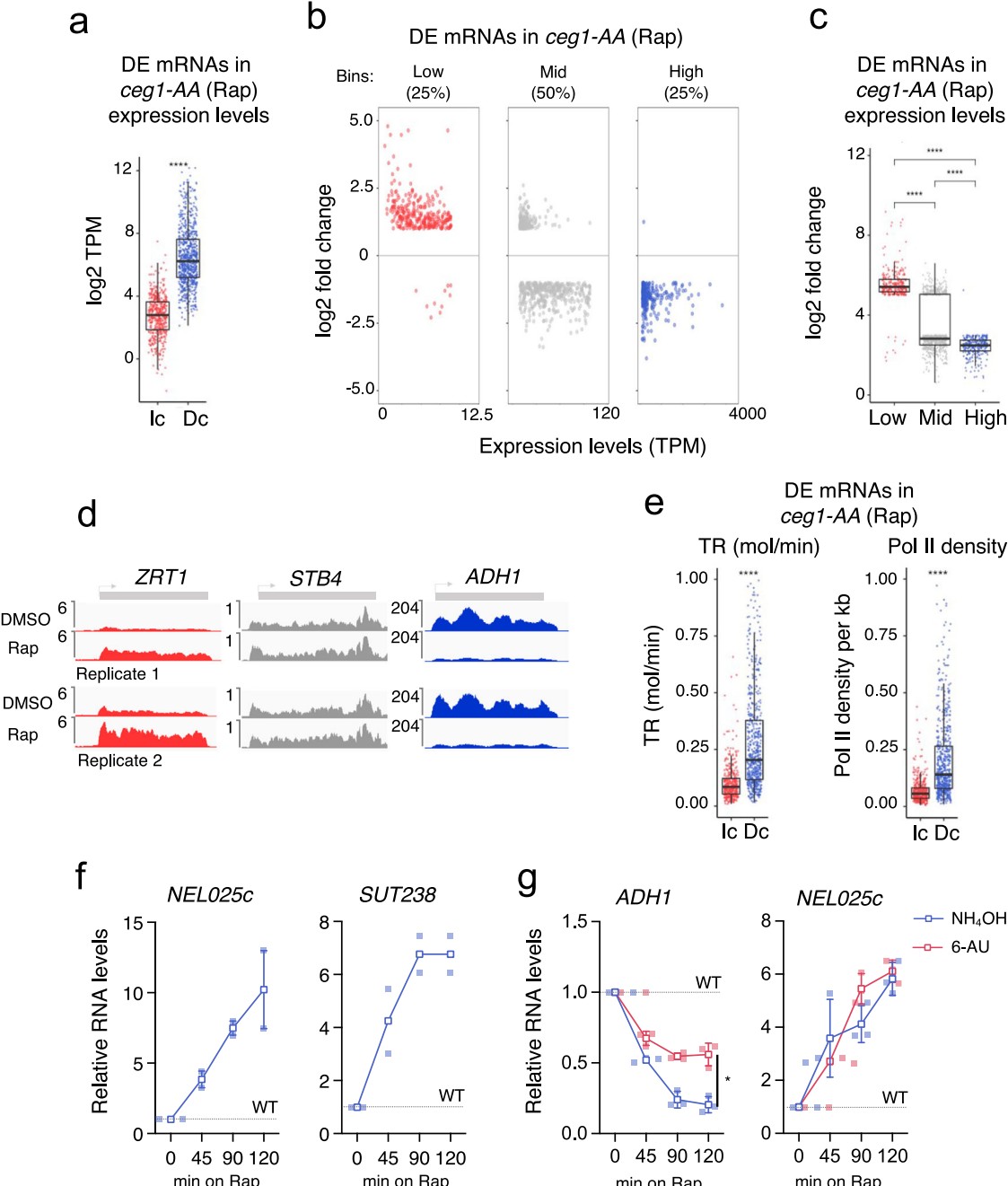

**Fig. 2 The accumulation of non-capped RNAs depends on gene expression levels. a** Differentially expressed mRNAs in *ceg1-AA* (Rap) vs *ceg1-AA* (DMSO/WT) sorted by their expression levels: Increased (Ic) or decreased (Dc). The expression levels were calculated by the log2 of Transcript Per Million (log2 TPM) value in WT (*ceg1-AA* on DMSO). The asterisks (*) indicate the statistical significance calculated via *t* test. **b** The distribution of differentially expressed mRNAs in *ceg1-AA* (Rap) depending on their basal expression levels in WT cells. **c** The overall fold change for differentially expressed mRNAs in *ceg1-AA* (Rap) vs *ceg1-AA* (DMSO) ranked according to their basal expression levels. The asterisks (*) indicate the statistical significance calculated via ANOVA. **d** Genome viewer tracks showing RNA-seq reads in *ceg1-AA* (DMSO) and *ceg1-AA* (Rap) for lowly (red), medial (grey) and highly (blue) expressed genes. **e** Box plots showing transcription rates and Pol II density for differentially expressed mRNAs in *ceg1-AA* (Rap) vs *ceg1-AA* (DMSO). The asterisks (*) indicate the statistical significance calculated via *t* test. **f** The accumulation of the *NEL025c* (left) and *SUT238* (right) RNAs in *ceg1-AA* strain upon 0, 45, 90 and 120 min of rapamycin treatment compared to WT (DMSO) set to 1 (dotted line). Analysis of *SUT238* $n = 2$, individual points of biological replicates are shown. **g** RT-qPCR analysis showing the levels of *ADH1* mRNA and *NEL025* ncRNA upon 6-AU (red) and $NH_4OH$ (blue) treatments in Ceg1-depleted cells. The error bars show the standard deviation of three independent experiments. Statistical analysis is shown for the last time point (120 min). **a**, **c**, **e** ns = $P > 0.05$; * = $P \leq 0.05$; ** = $P \leq 0.01$; *** = $P \leq 0.001$; **** = $P \leq 0.0001$. The box limits show the first and the third quartile (Q1 and Q3). The line inside the box represents the median value. The lines (whisker) show the maximum and minimum value within 1.5 times interquartile Q1 and Q3. The minimum/maximum whisker values are calculated as Q1/Q3 ∓ 1.5 * IQR (interquartile range). Each point represents the value relative to a single gene.

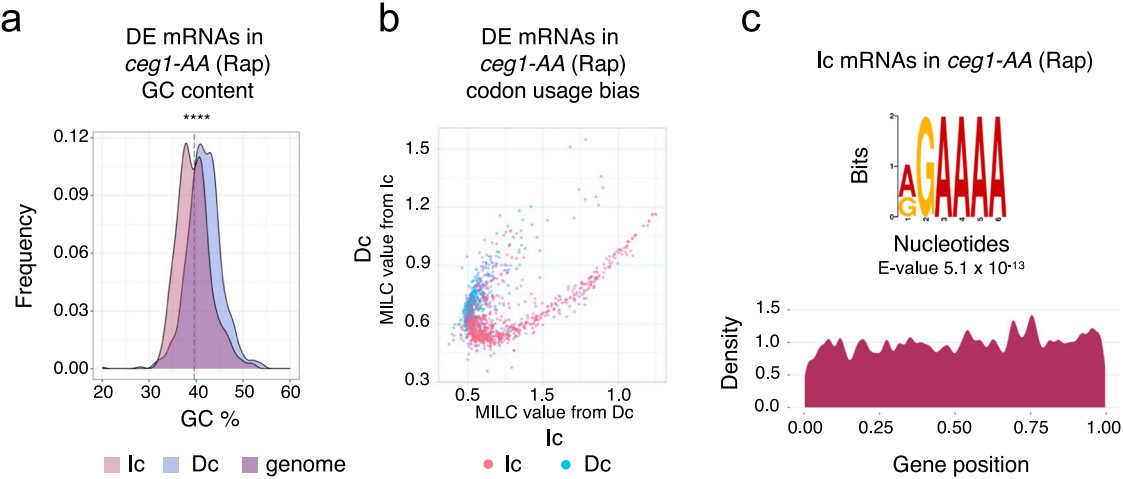

**Fig. 3 The stability of non-capped mRNAs is associated with differential features of coding sequences. a** The GC content of increased (Ic) and decreased (Dc) mRNAs upon Ceg1 nuclear depletion. The GC content percentage distribution frequency is shown as a density plot. The statistical significance was assessed via Chi-squared and Student's *t* test. ns = $P > 0.05$; * = $P \leq 0.05$; ** = $P \leq 0.01$; *** = $P \leq 0.001$; **** = $P \leq 0.0001$. The vertical dotted line denotes the median GC content of the yeast genome. **b** The bias in codon usage between Increased (Ic) and decreased (Dc) genes differentially expressed upon nucleodepletion of Ceg1 for 45 min. The plot shows the Measure Independent of Length and Composition (MILC) distance between the two sets of genes. **c** Logo sequence overrepresented in the increased mRNAs in *ceg1-AA* (Rap).

involved in the degradation of mRNAs with unmethylated caps[14]. Thus, we investigated if Rat1 inactivation could rescue the levels of non-capped mRNAs. We employed the same experimental system, *ceg1-63* and *ceg1-63/rat1-1* temperature-sensitive mutants, to assess the levels of highly expressed *ADH1* as well as *GAL1* and *FMP27* genes previously used to study capping-dependent RNA degradation[20]. *FMP27* is a lowly expressed gene placed under control of the strong inducible *GAL1* promoter in the *ceg1-63* strain[20]. We shifted the mutants and the parental strain to a non-permissive temperature (37 °C) for up to 120 min, for the analysis of *GAL1::FMP27* the strain grew on galactose-containing medium. The mRNA levels of both *ADH1* and *GAL1::FMP27* decreased in *ceg1-63* however, the inactivation of Rat1 in *ceg1-63/rat1-1* did not rescue their abundance (Fig. 4a). The profile of the *FMP27* levels expressed from the *GAL1* promoter were mirrored by the levels of endogenous *GAL1* mRNA (Fig. 4a) confirming that the mRNA levels were correlated with the strength of RNA synthesis in *ceg1-63*. Notably, *FMP27* is expressed at a low level from its own promoter in WT cells and did not decrease in *ceg1-AA* but remained stable (Fig. 4b). We did not observe that mRNA levels in *ceg1-63* were rescued by Rat1 inactivation at sub-permissive temperature (34 °C) (Supplementary Fig. 4a). Similar to mRNAs, the levels of lowly transcribed ncRNA *NEL025c* were not altered by the *rat1-1* mutation in the *ceg1-63* strain (Fig. 4a and Supplementary Fig. 4a). Although Rat1 may moderately delay degradation of non-capped mRNA[20] our data indicate that Rat1 has rather minor role in the overall clearing of non-capped mRNAs.

To avoid metabolism-changing temperature shifts of *ceg1-63/rat1-1* we tested how the deletion of the gene encoding the Rat1 binding partner Rai1 impacts RNA levels in *ceg1-AA*. Rai1 converts triphosphorylated RNA 5′ ends generated by Pol II into monophosphate 5′ RNA which are substrates for 5′−3′ exonucleases[44, 45]. Rai1 also stabilises and activates Rat1 by forming a stable Rai1-Rat1 complex and together with Npl3 control degradation of cap-defective mRNAs[15,46–48]. The Ceg1 nuclear depletion resulted in a decrease of *ADH1* mRNA in strains with either functional or deleted *RAI1* (Fig. 4c). This further confirmed that Rat1 may not be efficiently involved in the degradation of non-capped transcripts.

Previous studies reported that shifting the temperature-sensitive mutant *ceg1-63* to sub-permissive temperature resulted

in a premature transcription termination of the eight kb-long gene *FMP27* via a mechanism associated with Rat1-dependent co-transcriptional degradation of nascent RNA[20]. However, this effect was not fully recapitulated in the *ceg1-AA* strain consistently with our data indicating that Rat1 is not efficiently involved in degradation of non-capped mRNAs. ChIP analysis using an antibody recognising Pol II CTD revealed only a modest reduction of Pol II occupancy ~0.5 kb from the transcription start site but not further downstream of the gene (Fig. 4d). Since the average gene length in yeast is 1.4 kb[49] (Supplementary Fig. 4b), we investigated how transcription was affected in *ceg1-AA* on shorter genes.

We analysed the distribution of Pol II phosphoisoforms on highly and lowly expressed genes *ADH1* and *PML39*, respectively, in *ceg1-AA* (Fig. 4e, f). Total Pol II was partially depleted mainly on the 5′ ends for both highly and lowly expressed genes. Since we used 8WG16 antibody recognising non-modified heptapep-tides of the CTD, we speculate that this decrease of the Pol II signal may reflect prematurely terminated unphosphorylated or hypophosphorylated Pol II. Indeed, both ser5-P and ser2-P Pol II phosphoisoforms, which represent actively transcribing Pol II, were only marginally affected by Ceg1 depletion (Fig. 4e, f). A similar but less pronounced effect was observed for the *ADH1* gene in *cet1-AA* (Supplementary Fig. 4c). Nuclear depletion of Abd1 did not affect the distribution of Pol II over *ADH1* (Supplementary Fig. 4d). We also investigated if longer nuclear depletion of Ceg1 resulted in increased premature transcription termination. We shifted cells to the medium containing Rap for up to 120 min and tested the levels of Pol II over the 3′ end of high and mid-range expressed *ADH1*, *PYK2* and *PMA1* genes (Fig. 4g). Consequently, we observed only ~20% decrease in Pol II signal at 45 min which was not exacerbated by longer nuclear depletion of Ceg1.

**XRN1 deletion reciprocates Ceg1 nuclear depletion on the RNA level.** Next, we tested if *ADH1* levels could be rescued in *ceg1-AA* by the inactivation of the other major 5′−3′ exonuclease, Xrn1. Indeed, the deletion of *XRN1* restored the levels of the non-capped *ADH1* mRNA in *ceg1-AA* (Fig. 5a). Note that the *ceg1-AA/xrn1Δ* strain was non-viable on Rap-containing medium (Fig. 5b) confirming that non-capped mRNAs were not

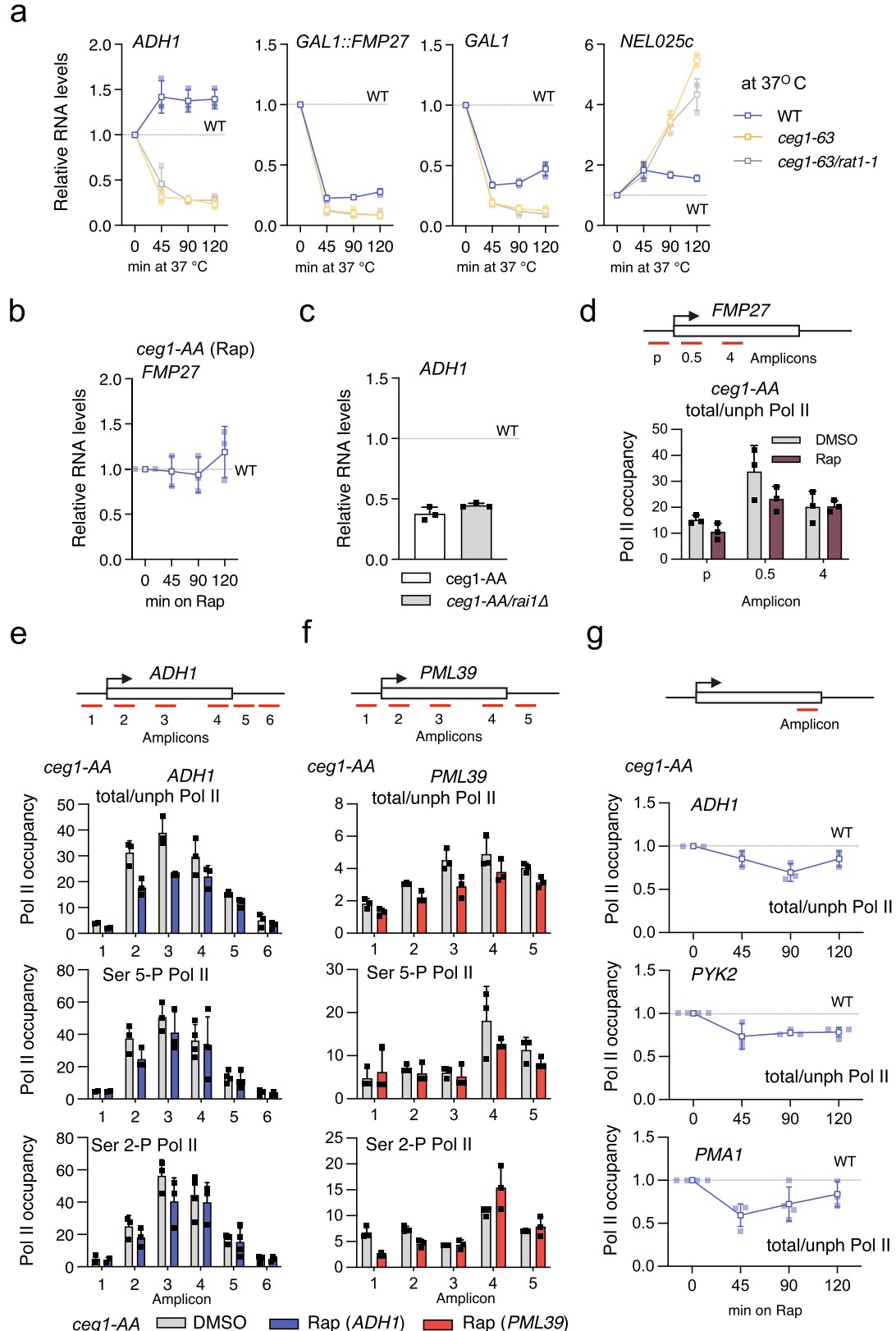

functional in this strain. The rescue effect observed for *ADH1* in *ceg1-AA/xrn1Δ* (Rap) prompted us to test whether the differential expression in *ceg1-AA* could be restored globally by the deletion of *XRN1*. We performed RNA-seq analysis on the *ceg1-AA/xrn1Δ* strain growing on either DMSO (*xrn1Δ*) or Rap for 45 min and found only 18 decreased non-capped mRNAs, 13 of which were

also reduced in *ceg1-AA* (Fig. 5c, e). Surprisingly, the number of increased mRNAs was also significantly lower in *ceg1-AA/xrn1Δ* when compared to *ceg1-AA* as we detected only 23 upregulated mRNAs (Fig. 5c, d). The observation that *XRN1* deletion rescues increased mRNAs in *ceg1-AA* was somewhat unexpected. To elucidate this, we tested mRNA expression in *ceg1-AA/xrn1Δ*

**Fig. 4 Participation of the 5′−3′ exonuclease Rat1 in the degradation of non-capped transcripts. a** RNA levels of *ADH1*, *FMP27*, *GAL1* and *NEL025c* in WT and temperature-sensitive mutants *ceg1-63* and *ceg1-63/rat1-1* after 0, 45, 90 and 120 min at a non-permissive temperature (37 °C) compared to control (WT at 25 °C) set to 1 (dotted lines). For analysis of *GAL1:FMP27* and *GAL1* mRNAs cells grew on galactose-containing medium. **b** *FMP27* mRNA levels in *ceg1-AA* after 0, 45, 90 and 120 min of rapamycin treatment compared to WT (*ceg1-AA* on DMSO) set to 1 (dotted line). **c** *ADH1* mRNA levels in *ceg1-AA* and *ceg1-AA/rai1Δ* after 45 min on Rap compared to isogenic controls growing on DMSO set to 1 (dotted line). **d** The distribution of total/unphosphorylated Pol II (total/unph Pol II) over *FMP27* in control (DMSO) and after 45 min of rapamycin treatment (Rap) in *ceg1-AA*. The location of the amplicons used for the ChIP-qPCR over *FMP27* is shown above the chart. The error bars show the standard deviation of three independent ChIP-qPCR experiments. The distribution of total/unphosphorylated Pol II (total/unph Pol II) as well as serine 5-phosphorylated (Ser5-P Pol II) and serine 2-phosphorylated (Ser2-P Pol II) Pol II isoforms over *ADH1* (**e**) and *PML39* (**f**) in control (DMSO) and after 45 min of rapamycin (Rap) in *ceg1-AA* strain. The location of the amplicons used for the ChIP-qPCR is shown above the charts. The error bars show the standard deviation of three independent ChIP-qPCR experiments. **g** The occupancy of the total/unphosphorylated Pol II at the 3′ end of *ADH1*, *PYK2* and *PMA1*. Timepoints after 0, 45, 90 and 120 min on rapamycin compared to control (*ceg1-AA* on DMSO) and set to 1 (dotted line). The location of the amplicons used for the ChIP-qPCR is shown above the charts. The error bars show the standard deviation of three independent ChIP-qPCR experiments.

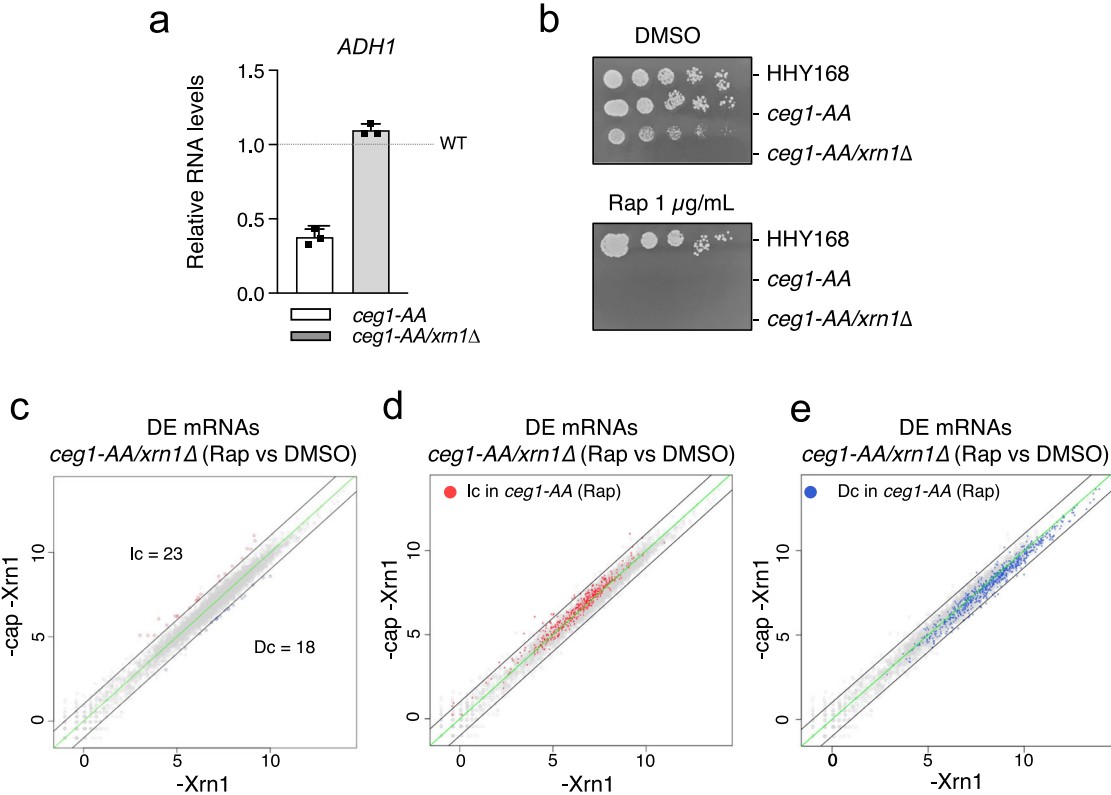

**Fig. 5 The deletion of the 5′−3′ exonuclease Xrn1 rescues the levels of non-capped transcripts. a** *ADH1* mRNA levels in *ceg1-AA* and *ceg1-AA/xrn1Δ* after 45 min on Rap compared to isogenic controls growing on DMSO set to 1 (dotted line). RT-qPCR, the error bars show the standard deviation of three independent experiments. **b** Spot test showing the growth of *ceg1-AA/xrn1Δ* on DMSO and rapamycin. Serial dilutions of each strain were spotted on YPD media containing DMSO or rapamycin (1 μg/ml). **c** Differential expression of mRNAs after 45 min of rapamycin treatment in *ceg1-AA/xrn1Δ* (-cap -Xrn1) compared to the control condition *ceg1-AA/xrn1Δ* on DMSO (-Xrn1). Decreased (Dc) RNA species showing log2 fold change <-1 are in blue. Increased (Ic) RNA species with log2 fold change >1 are labelled in red. The zero-change line is in green. The black lines represent the ±1 log2 fold change threshold. The average values with padj < 0.05 of *n* = 2 experiments are shown. Increased (Ic) (**d**) or decreased (Dc) (**e**) mRNAs in *ceg1-AA* (Rap) are labelled on the plot showing mRNA levels in *ceg1-AA/xrn1Δ* (Rap) compared to the control condition *ceg1-AA/xrn1Δ* (DMSO). The average values with padj < 0.05 of *n* = 2 experiments are shown.

prior to the nuclear depletion of Ceg1. We compared *ceg1-AA/xrn1Δ* (DMSO) to WT (*ceg1-AA* on DMSO) and found that *XRN1* deletion results in a global change in the mRNA expression which was similar to the pattern observed upon Ceg1 nuclear depletion. RNA-seq analysis of *ceg1-AA/xrn1Δ* on DMSO (referred as *xrn1Δ*) revealed 543 upregulated and 605 down-regulated mRNAs with log2 fold change <−1 and >1 (Fig. 6a). The upregulated genes were lowly expressed (median log2 TPM = 2.6) while the decreased fraction was highly expressed (median log2 TPM = 6.2) in WT cells (Fig. 6b). Similarly to the previous analysis, all of 1148 mRNAs differentially expressed in

*xrn1Δ* were split into three bins according to their expression levels in WT cells (Fig. 6c, d). The majority of the lowly expressed mRNAs increased (270 out of 282), while most of the highly expressed mRNAs decreased (280 out of 282) in *xrn1Δ*. The number of mid-range expressed genes was equally affected (268 upregulated, 296 downregulated) however, the median of their expression levels was overall decreased. Consistently, upregulated genes displayed lower transcription rates and Pol II density than downregulated genes (Fig. 6e). Since the observation in *xrn1Δ* very closely resembles those in *ceg1-AA* on Rap, we tested if the effect of nuclear depletion of Ceg1 in *xrn1Δ* strain was masked by

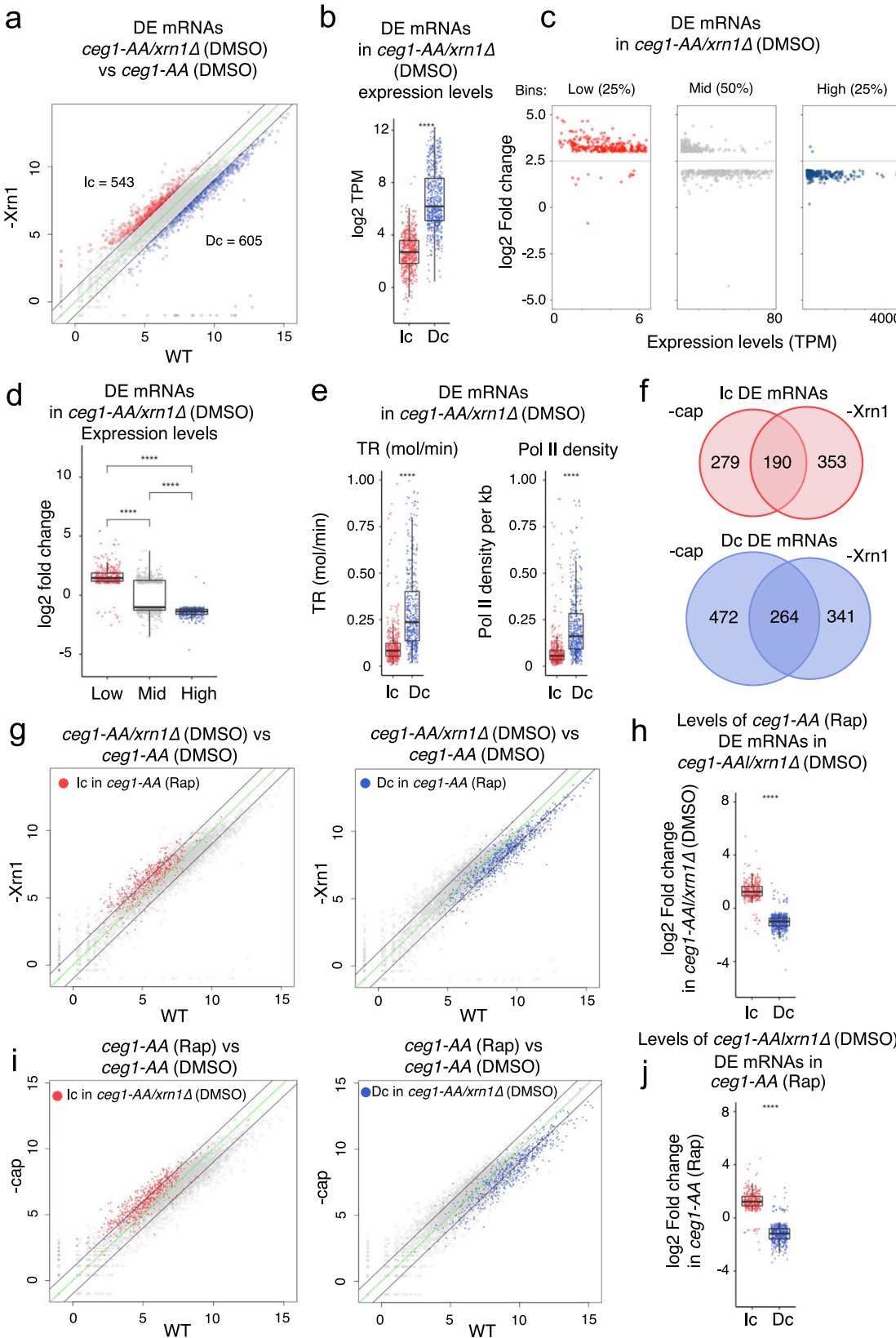

the pre-existing aberrant accumulation of mRNAs caused by the *XRN1* deletion. We assessed how many differentially expressed mRNAs in *ceg1-AA* were also differentially expressed in *xrn1Δ* prior to nuclear depletion of Ceg1. We found that 190 out of 469 mRNAs increased and 264 out of 736 mRNAs decreased in *ceg1-AA* were also similarly affected in *xrn1Δ* (Fig. 6f). This overlap

identified only genes with log2 fold change over the ±1 threshold in both data sets therefore, we also assessed the levels of mRNAs differentially expressed in *ceg1-AA* in the total mRNA fraction in *xrn1Δ*. Generally, most of the increased and decreased mRNAs identified in *ceg1-AA* were similarly affected in *xrn1Δ* (Fig. 6g) displaying a mean log2 fold change of 1.28 for the increased and

**Fig. 6 XRN1 deletion and Ceg1 nuclear depletion similarly affect mRNA levels. a** Differential levels of mRNAs in *xrn1Δ* (*ceg1-AA/xrn1Δ* on DMSO) compared to WT (*ceg1-AA* on DMSO). Increased (Ic) mRNAs with log2 fold change >1 are labelled in red, decreased (Dc) mRNAs showing log2 fold change <-1 are in blue. The average values with padj < 0.05 of *n* = 2 experiments are shown, the zero-change line is in green. The black lines represent the ±1 log2 fold change threshold. **b** Differentially expressed mRNAs in *xrn1Δ* (*ceg1-AA/xrn1Δ* on DMSO) sorted by their expression levels calculated by the log2 Transcript Per Million (log2 TPM) value in WT (*ceg1-AA* on DMSO). **c** The distribution of differentially expressed mRNAs in *xrn1Δ* depending on their expression levels in WT cells. **d** The overall fold change for differentially expressed mRNAs in *xrn1Δ* ranked according to their expression levels in WT cells. **e** Transcription rates and Pol II density for differentially expressed mRNAs in *xrn1Δ* (*ceg1-AA/xrn1Δ* on DMSO). **f** The number of overlapping increased (Ic) or decreased (Dc) mRNAs in *xrn1Δ* (*ceg1-AA/xrn1Δ* on DMSO) and *ceg1-AA* (Rap). **g** Increased (red) or decreased (blue) mRNA in *ceg1-AA* (Rap) are labelled on the plots showing mRNA levels in *ceg1-AA/xrn1Δ* (DMSO) compared to the control condition (WT). The average values with padj <0.05 of *n* = 2 experiments are shown. **h** The log2 fold change in *xrn1Δ* for the sets of mRNAs differentially expressed in *ceg1-AA* (Rap). **i** Increased (red) or decreased (blue) mRNAs in *xrn1Δ* are labelled on the plots showing mRNA levels in *ceg1-AA* (Rap) compared to the control condition (WT). The average values with padj <0.05 of *n* = 2 experiments are shown. **j** The log2 fold change in *ceg1-AA* (Rap) for the sets of mRNAs differentially expressed in *xrn1Δ*. **b, d, e, h, j** The asterisks (*) indicate the statistical significance calculated via t-test or ANOVA. ns = $P > 0.05$; * = $P \leq 0.05$; ** = $P \leq 0.01$; *** = $P \leq 0.001$; **** = $P \leq 0.0001$. The box limits show the first and the third quartile (Q1 and Q3). The line inside the box represents the median value. The lines (whisker) show the maximum and minimum value within 1.5 times interquartile Q1 and Q3. The minimum/maximum whisker values are calculated as Q1/Q3 ∓ 1.5 * IQR (interquartile range). Each point represents the value relative to a single gene.

−1.01 for the decreased mRNAs (Fig. 6h). We also performed reverse analysis and tested the levels of differentially expressed mRNA in *xrn1Δ* in *ceg1-AA* (Fig. 6i). Both groups are accordingly increased or decreased in *ceg1-AA* by a mean log2 fold change of 1.2 and −1.1, respectively (Fig. 6j). These analyses indicate that *XRN1* deletion and nuclear depletion of capping enzymes have very similar effect on mRNA accumulation. This suggests a common mechanism affecting mRNA levels and therefore explains why the Xrn1 and Ceg1 co-inactivation did not result in many differentially expressed mRNAs. We concluded that the lack of the cap at the 5′ ends of mRNAs is insufficient to initiate Xrn1-dependent degradation and the process of the cap removal is the key step marking mRNAs for 5′−3′ degradation. Finally, we compared the RNA-seq dataset from *ceg1-AA* (Rap) and *ceg1AA/xrn1Δ* (Rap) to exclude the genes affected by either *XRN1* deletion or Ceg1 depletion. We identified 482 up- and 536 down-regulated (Supplementary Fig. 5a), both groups are expressed on similar levels in WT cells (Supplementary Fig. 5b).

**Removal of the cap stimulates mRNA degradation of lowly transcribed genes.** The similar patterns of mRNA levels in *ceg1-AA* and *xrn1Δ* strains imply that decapping is required for 5′−3′ Xrn1-dependent regulation of mRNA levels. Non-capped mRNAs may not be immediately subjected to efficient degradation upon Ceg1 depletion as they are not direct substrates for the 5′−3′ exonucleases and require modifications of the 5′ ends by other enzymes[44,45]. Thus, we anticipated that even lowly expressed non-capped mRNAs with a 5′ end suitable for the 5′−3′ exonucleases would be efficiently removed and would not accumulate as observed following Ceg1 nuclear depletion. To test this hypothesis, we introduced the self-cleaving hepatitis delta virus (HDV) ribozyme sequence into the 5′ UTR of the *FMP27* gene (*RZ-fmp27*) (Fig. 7a) in the *ceg1-AA* strain. The internal self-cleavage in the ribozyme sequence generated a non-capped transcript *RZ-fmp27* with a 5′-OH end which is a suitable substrate for 5′−3′ exonucleases[44,45]. As a control, we used the catalytically inactive ribozyme mutant harbouring the substitution C76:U (*C76U-fmp27*) preventing the cleavage[50,51]. Both forms were expressed from the *ADH1* promoter which increased *C76U-fmp27* levels ~6 times compared to the WT *FMP27* level (Fig. 7b) however, this expression was still magnitudes lower than *ADH1* levels (Fig. 7c), retaining *C76U-fmp27* in the range of lowly expressed genes. Our analysis showed that lowly expressed *FMP27* mRNA did not change when synthesised as non-capped RNA in *ceg1-AA* (Fig. 4b). Similarly, *C76U-fmp27* level was not affected by Ceg1 nuclear depletion (Fig. 7d). However, the functional ribozyme generating a non-capped transcript decreased the levels of *RZ-fmp27* by 67%

compared to the catalytically inactive *C76U-fmp27* on DMSO (WT condition) and by 70% during Ceg1 nuclear depletion (Fig. 7d). This indicates that *RZ-fmp27* mRNA decapped by the self-cleaving ribozyme, which produced a 5′ end suitable for 5′−3′ exonucleases, was efficiently degraded. This suggests that the generation of the correct substrate was required for the 5′ quality control as previously reported[46].

## Discussion

Capping is the first co-transcriptional RNA modification imposed on mRNAs. The m[7]G cap is essential for the functionality of the mRNA and therefore, mutations affecting the cap formation are non-viable. Here, we performed a global analysis of the cap defective transcriptome. We show that although the decrease of mRNA levels was a dominant effect caused by the nuclear depletion of the capping enzymes, many mRNAs increased when not capped. This seems to contrast the previous studies reporting that selected RNAs with defective caps (e.g., lacking cap or with non-methylated caps) are rapidly degraded by 5′−3′ exonucleases[1, 14,20]. These reports indicated the role of Rai1 and Rat1 in degradation of capped but not methylated mRNAs[14] and showed that Rat1 mutation initially slowed down degradation of non-capped *FMP27* and *GAL1* mRNAs but overall did not rescue their levels in *ceg1-63* mutant[20]. Our genome-wide data show a clear dependence between the expression strength and the degradation of uncapped transcripts. The nuclear depletion of Ceg1 resulted in the reduction of highly expressed mRNAs and the accumulation of lowly expressed and more stable mRNAs. We observed the same pattern in cells lacking the 5′−3′ exonuclease Xrn1 which is one the major enzymes degrading mRNA in the cytoplasm. Consistently, we did not detect any significant changes in RNA levels in the *xrn1Δ ceg1-AA* strain when compared to the single *ceg1-AA* mutant. Our data indicate that the presence of cap is crucial to initiate 5′−3′ exonucleolytic degradation and is required for the Xrn1-dependent control and buffering of the global mRNA levels[22]. Indeed, a subunit of the decapping complex Dcp1-Dcp2, Dcp1 interacts with Xrn1, and the cap removal is the first step triggering mRNA degradation[21,52]. Thus, mRNA synthesised as non-capped transcripts may not be efficiently degraded by 5′−3′ exonucleases like Xrn1 or Rat1. In such a scenario, the highly expressed non-capped RNAs decreases significantly as they may be preferentially degraded by the 3′−5′ degrading machinery involving dead-enylases and the exosome complex[40]. This in turn, may increase the relative concentration of lowly transcribed mRNAs in the overall RNA fraction since their degradation may rely mainly on 5′−3′ pathway[40] (Fig. 8). We speculate that the binding of the

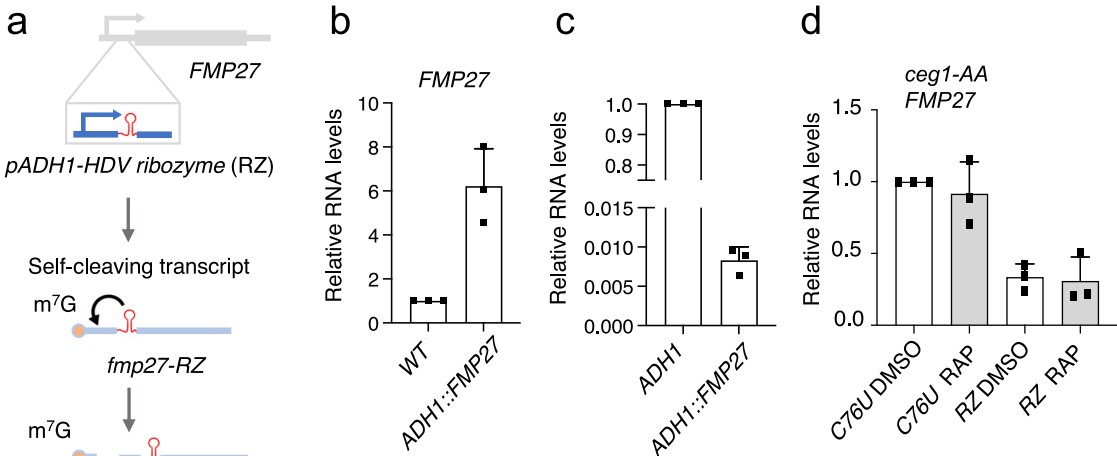

**Fig. 7 Ribozyme-dependent cleavage induces *FMP27* mRNA degradation. a** The experimental approach for the generation of the self-cleaving *RZ-fmp27* transcript. Comparison of *C76U-fmp27* mRNA levels with WT *FMP27* (**b**) and *ADH1* (**c**) mRNAs. **d** mRNA levels of *RZ-fmp27* and *C76U-fmp27* in *ceg1-AA* growing on Rap or DMSO. **b**, **d** The error bars show the standard deviation of three independent RT-qPCR experiments.

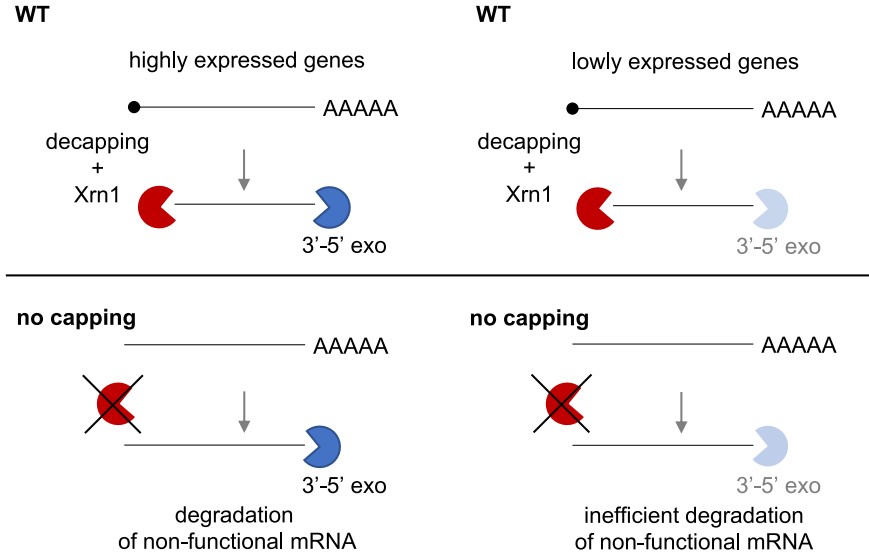

**Fig. 8 The model depicting class-specific degradation of non-capped mRNAs.** 5′−3′ degradation is affected by the lack of the 5′ end capping during mRNA synthesis. Non-capped highly expressed mRNAs are degraded by 3′−5′ exonucleases while lowly expressed mRNAs accumulate as their turnover relies mainly on the Xrn1 exonuclease.

capping enzymes to the transcriptional machinery or/and the initiation of m7G cap synthesis, is required to activate the cap quality control pathway. Such a condition would allow for the detection and removal of mRNA with defective caps[15] and also aid the production of other essential non-capped but stable RNA classes such as ribosomal RNA (rRNA) synthesised by Pol I[7,53] whose processing and termination employs Rat1[54,55]. Indeed, the capping process plays a role in marking transcripts as mRNAs. For example, yeast box C/D snoRNAs, a class of Pol II transcripts which do not possess the m7G cap as mature species, are subjected to mRNA processing pathways if the caps are not co-transcriptionally removed from their precursors[56].

The m7G cap contributes but may not be essential for the protection from exonucleolytic 5′−3′ degradation, as the synthesised 5′ triphosphate non-capped mRNAs are not direct substrates of the 5′−3′ exonucleases[44, 45]. Indeed, the generation of the 5′ end suitable for 5′−3′ exonucleases by the insertion of the self-cleaving HDV ribozyme into the 5′ UTR of *FMP27* resulted in enhanced degradation of this lowly transcribed mRNA in the WT strain, confirming that the formation of a correct substrate is

crucial for the 5′−3′ degradation machinery. Similarly, Rat1/Xrn1-dependent 5′−3′ processing of yeast snoRNAs depends on capping removal by decapping enzymes or endonuclease Rnt1[57].

Finally, we did not observe a significant increase in premature transcription termination upon depletion of the capping enzymes. This is consistent with our data showing that Rat1 and Rai1 inactivation did not restore the levels of non-capped mRNAs, but in contrasts with previous models suggesting that such RNAs are degraded co-transcriptionally by Rat1 eliciting transcription termination[16–18]. Our ChIP analysis revealed that only a small amount of Pol II tested using 8WG16 antibody was depleted from the chromatin in *ceg1-AA*. This antibody recognises unphosphorylated CTD and therefore the precipitated fraction may be enriched for hypophosphorylated Pol II. Such Pol II may not efficiently transit to the elongation phase, pause or transcribe at a low speed[58–60]. All these conditions facilitate transcription termination, which may be reflected by the lowered levels of Pol II observed in the ChIP analyses employing 8WG16 antibody. This effect was not detected when the distribution of Pol II was measured using antibodies recognising actively transcribing complexes. We speculate that for average length noncapped mRNAs

(1.4 kb), the opportunity window is too narrow for Rai1-Rat1 to be recruited to the transcribing complex, modify the 5′ end and initiate co-transcriptional degradation and so to trigger transcription termination. Such premature transcription termination is observed when nascent RNAs are decapped which enforces Pol II pausing facilitating Rat1-dependent termination[18,19,61,62].

Overall, our work reveals the complexity of the 5′ cap functions in cellular mRNA metabolism. The action of the capping enzyme during transcription and then the presence of the cap structure on the mature mRNA as well as the process of its removal generating an mRNA substrate for 5′−3′ exonucleases may all be crucial for the degradation of mRNA transcribed on low levels in the cell. Overall, cap may play an essential role in marking mRNA molecules and cooperating with Xrn1-dependent degradation and regulation of the cellular mRNA levels.

## Methods

**Oligonucleotides used in the study**. Oligonucleotides used for strain construction, RT-qPCRs and ChIP analyses are listed in Supplementary Data 3.

**Yeast strain and media**. Strains: HHY168 (MATalpha tor1-1 fpr1::NAT RPL13AxFKB12::TRP), Euroscarf; *ceg1-AA* (HHY168 CEG1-FRB::HIS), this work; *ceg1-GFP-AA* (HHY168 CEG1-GFP-FRB::HIS), this work; *cet1-AA* (HHY168 CET1-FRB::HIS), this work; *abd1-AA* (HHY168 ABD1-FRB::HIS), this work; *cbp20-AA* (HHY168 CBP20-FRB::HIS); this work; *cbp20-GFP-AA* (HHY168 CBP20-GFP-FRB::KAN), this work; *cbp80-AA* (HHY168 CBP80-FRB::HIS), this work; *cbp80-GFP-AA* (HHY168 CBP80-GFP-FRB::KAN), this work; *ceg1-AA/xrn1Δ* (HHY168 CEG1-FRB::HIS KAN::XRN1), this work; *ceg1-AA/rrp6Δ* (HHY168 CEG1-FRB::HIS KAN::RRP6), this work; *ceg1-63* (MATa ceg1Δ::HIS3 PGAL1-YLR454::URA3 ura3Δ [pRS315-ceg1-63])[20]; *ceg1-63/rat1-1* (MATa rat1-1 ceg1Δ::HIS3 PGAL1-YLR454::URA3 ura3Δ [pRS315-ceg1-63])[20]; *fmp27-C76U* (HHY168 CEG1-FRB::HIS URA3::RZc76u::YLR454), this work; *fmp27-RZ* (HHY168 CEG1-FRB::HIS URA3::RZ::YLR454), this work; *ceg1-AA/rai1Δ* (HHY168 CEG1-FRB::HIS KAN::RAI1), this work.

Yeast strains were grown at 30 °C on YPD (1% yeast extract, 2% peptone, 2% dextrose/glucose) media enriched in DMSO or rapamycin to the final concentration of 1 µg/mL where indicated. For 6-AU treatment, the yeast cells were grown in minimal media (yeast nitrogen base without amino acids) with synthetic dropout ura-. For solid media, agar was added to the final concentration of 2%.

**Growth spot assay**. Yeast cells were grown to OD600 = 0.4. Serial dilutions 1:25 were performed and 5 µL of each dilution was spotted onto YPD (2% agar) plates.

**Yeast cell transformation**. Cells were cultured to OD600 = 1 and pelleted by centrifugation for 5 min at 2400 × g at room temperature (RT). The pellet was resuspended in sterile 10 mM Tris-HCl pH 7.5 or water and centrifuged again as before. The pellet was resuspended in filtered LiT (10 mM Tris-HCl pH 7.5;100 mM Lithium acetate), 1 M DTT and incubated at RT for 40 min with gentle shaking. After incubation the cells were pelleted again (as before) and resuspended in LiT and 1 M DTT. The suspension was added with LiT; and dsDNA (10 mg/mL); transforming DNA (0.1 to 1 µg or more) and incubated at RT for 10 min. Next, PEG solution (1:1 PEG4000:LiT) was added to the suspension and incubated at RT for 10 min. DMSO was then added, and the suspension incubated at 42 °C for 15 min. After the incubation, the suspension was pelleted and resuspended in 1 mL of YPD (1% Bacto yeast extract, 2% Bacto peptone, 2% glucose/dextrose). Then, the suspension was incubated at 30 °C for 1 h, pelleted with 10 s spin, max speed and inoculated on selective plates.

**RNA extraction**. RNA was purified by phenol/chloroform method. The yeasts were cultured to the final concentration OD600 = 1, pelleted in 50 mL falcon tubes by centrifugation for 2 min at 1000 rpm at RT. The pellet was frozen at −80 °C before further manipulation. Next, the pellet was resuspended in 1 mL of ice-cold RNase-free water, moved to a 1.5 mL tube, centrifuged for 10 s at 4 °C max speed. Then the supernatant (water) was removed, and the pellet was resuspended in 400 µL of filtered AE buffer (50 mM Sodium acetate pH5.3; 10 mM EDTA), 40 µL of 10% Sodium dodecyl sulphate (SDS) and 400 µL of Acid Phenol (pH 4). The suspension was vortexed for 20 s and incubated at 65 °C for 10 min, then incubated for further 10 min at −80 °C. The defrosted sample was then centrifuged for 5 min, 13,000 rpm at RT. 400 µL of 1:1 phenol:chloroform solution were added to the supernatant and vortexed for 30 s. Then the sample was centrifuged 10 min, 13,000 rpm at RT. The upper phase was transferred and mixed with 400 µL of Chloroform, vortexed and centrifuged for 5 min, 13,000 rpm at RT. The RNA was precipitated by adding to the upper phase 1 mL of 100% ethanol and 40 µL of 7.5 M Ammonium acetate. Left at −80 °C for 2 h then centrifuged for 20 min, 13,000 at 4 °C. The pellet was washed with 70% ethanol (in water) and resuspended in RNase-free water.

**RT-qPCR**. Reverse transcription was performed with Super Script III Reverse transcriptase (Thermo Fisher, 18080044) following manufacturer's instruction using random hexamers (Thermo Fisher, N8080127) for cDNA synthesis. qPCR data were analysed with the ΔΔCt method normalised to 25S rRNA and control condition. The qPCR source data are listed in Supplementary Data 2.

**RNA sequencing**. Before library preparation, rRNA depletion was performed on 5 µg of total yeast RNA using 200 pmol of 3′ biotinylated probes (IDT) designed in house following the RiboPOP method[63]. The libraries preparation was performed with Ultra II RNA Library Prep Kit for Illumina (NEB, E7770) and Multiplex Oligos for Illumina (NEB, E7335) according to manufacturer's instructions. The RNA sequencing was performed from the Genomics Facility at the University of Birmingham on Illumina NEXTseq apparatus.

**RNA-seq data analysis**. The quality of the data was checked using FastQC v0.11.5 (Andrews S. (2010). FastQC: a quality control tool for high throughput sequence data. Available online at: http://www.bioinformatics.babraham.ac.uk/projects/fastqc).

Reads were aligned to the yeast genome R64 (sacCer3) using HISAT2 v2.1.1 and sorted and indexed using SAMtools v1.15.1[64]. The read count per gene was calculated with LiBiNorm v.2.5[65] with the following parameters count -u name-root -f --order=pos --minaqual=10 --mode=intersection-strict --idattr=gene_id --type=exon. Differential expression analysis was performed using DESeq2 package v 4.3.1[66]

The GC content relative to the differentially expressed genes was calculated via EMBOSS infoseq (www.bioinformatics.nl/cgi-bin/emboss/infoseq). Increased (Ic) and decreased (Dc) genes were selected according to the results obtained from differential expression analysis performed via DESeq2 as previously described and their sequences interrogated for the distribution of GC content. The codon bias analysis was performed employing the cordon R package[67]; R package version 1.18.0, https://github.com/BioinfoHR/coRdon).

**Cap immunoprecipitation**. Total RNA from cells treated with DMSO or rapamycin was extracted and used for cap pull-down. 5 µg of total yeast RNA was spiked with 1 µg of total human RNA as immunoprecipitation efficiency control. Capped RNA species

were immunoprecipitated using 5 µg of anti m[7]G cap antibody (H-20) as previously described[20].

**6-azauracil (6-AU) treatment**. Overnight culture was prepared inoculating a single colony on minimal media as described above. The next day, the culture was diluted to OD600 = 0.2. The diluted culture was grown with agitation at 30 °C to OD600 = 1 and split in 4. Two cultures were added of 6-azauracil (6-AU) and the other two with ammonium hydroxide (NH4OH, control) at the final concentration of 50 µg/mL for 30 min. Next, DMSO or rapamycin (Rap) at the final concentration of 1 µg/mL were added according to the following scheme: 6-AU + DMSO, 6-AU +Rap, NH$_4$OH + DMSO, NH$_4$OH+Rap. 10 mL of each suspension were collected a time 0, 45, 90 and 120 min after the addition of DMSO or Rap and prepared for RNA purification.

**Chromatin immunoprecipitation (ChIP)**. The specific yeast strain was grown over-night to OD600 = 1. Then 37% formaldehyde was added to the media to final concentration 1% dropwise with gentle shaking for 15 min. Next, 15 mL of 2.5 M glycine was added for further 5 min keeping the shaking. The cells were pelleted by centrifugation at 3500 rpm for 3 min at 4 °C. The pellet was resuspended in 40 mL of ice-cold 1x PBS and centrifuged at 3500 rpm for 5 min at 4 °C, twice. The pellet obtained was then transferred and prepared for lysis. The pellet was resuspended in 1 mL of ice-cold FA1 (50 mM HEPES-KOH, pH7.5; 150 mM NaCl; 1 mM EDTA; 1% triton-x 100; 0.1% Sodium deoxycholate) added with proteinase inhibitor (Roche Mini-EDTA free) and about 400 µL zirconia beads. The cells were lysed in MagnaLyser (30 s 7000 rpm, 3 cycles with 5 min incubation on ice in between). The lysate was drained out of the beads by centrifugation 1000 rpm for 1 min at 4 °C and SDS was added to a final concentration of 0.05%. The lysate was sonicated as follows: 25 cycles 15 s ON, 15 s OFF on maximum power. Next, it was centrifuged for 20 min at 12,500 rpm at 4 °C. 200 µL of lysate was saved to check the fragmentation. SDS was added to the lysate with to final concentration of 1% and 0.05 mg of Proteinase K and incubated at 42 °C for 1 h. Next it was moved to 65 °C for 4 h. The DNA de-crosslinked was precipitated by Phenol/ Chloroform, the contaminant RNA was removed by adding 2 µL of RNase A incubating for 1 h at 37 °C and checked on agarose gel. The rest of the lysate was centrifuged at max speed for 5 min at 4 °C. 150 µg of chromatin were used for immunoprecipitation. FA1 enriched with proteinase inhibitor was added to the DNA to a final volume of 700 µL. 20 µL of such mix was saved as input for the further ChIP analysis. 2 µg of the specific antibody was added to the sample and incubated on rotation over-night at 4 °C.

15 µL of protein A (Thermo Fisher, 10002D) and 15 µL of protein G-coated beads (Thermo Fisher, 10004D) were washed with 1 mL of FA1 and resuspended in 100 µL of FA1. The beads were added to the sample and incubated by rotation for 1 h at 4 °C. After incubation the beads were washed six times with FA1, 1 time with FA2 (50 mM HEPES-KOH, pH 7.5; 500 mM NaCl; 1 mM EDTA; 1% Triton-X 100; 0.1% Sodium deoxycholate), one time with FA3 (20 mM Tris pH 8.0; 250 mM LiCl; 0.5% NP-40; 0.5% Sodium deoxycholate; 1 mM EDTA) and TE buffer pH 8 (100 mM Tris-HCl; 10 mM EDTA). The beads were then resuspended in 195 µL of Elution buffer (50 mM Tris pH 7.5; 10 mM EDTA; 1% SDS) and incubated for 10 min at 65 °C and 0.05 mg of Proteinase K added. The Chromatin was then de-crosslinked by incubating the sample at 42 °C for 1 h and moved at 65 °C for 4 h. The DNA was finally purified using Quiagen cleanup (28104) kit following the manufacturer's instruction. The ChIP-qPCR source data are listed in Supplementary Data 2.

**Microscopy of GFP-tagged strains**. Fixed cell microscopy was performed on yeast cells grown to late exponential phase in Synthetic Complete media with 2% glucose (SCD). Cells were treated with DMSO or rapamycin to a final concentration of 1 µg/ mL. 1 µL DAPI solution (2.5 mg/mL) was added to 1 mL treated yeast cells and vortexed gently before fixation in 100 µL 4% paraformaldehyde solution at room temperature for 15 min. 900 µL potassium phosphate/sorbitol (100 mM potassium phosphate pH 7.5, 1.2 M sorbitol) was added and cells centrifuged at top speed for 1 min, supernatant removed, and cells resuspended in 100 µL potassium sorbitol solution. 3 µL cells were mounted on 0.01% poly-L-lysine coated slides. Images were acquired using a Leica DM5500 fluorescent microscope. For quantification of the ratio between the mean signal intensity of the cytoplasm and nucleus, outlines were manually drawn using Image J for the whole cell and nucleus and the areas and mean signal intensities determined for each. The mean signal intensity of the cytoplasm was determined by calculating the total signal intensities, corrected against the background signal, and areas of the whole cell and the nucleus and subtracting that of the nucleus from cytoplasm followed by calculation of the ratio between mean cytoplasmic signal versus mean nuclear signal. Statistical significance was determined through one way ANOVA.

**Statistics and reproducibility**. The results presented depict the output of independent experiments ($n$ = 2–4). For each biological replicate shown, cells were grown, treated, and processed independently to minimise bias and validate reproducibility. The average values are shown, and the statistics performed via GraphPad Prism Version 10.0.2 represent the standard deviation resulting from the replicates analysed. For qPCR analysis where $n$ = 2, only the individual points are shown. Statistical significance was calculated via t-test for the comparison between two groups and ANOVA for comparison of >2 groups (ns = $P$ > 0.05; * = $P$ ≤ 0.05; ** = $P$ ≤ 0.01; *** = $P$ ≤ 0.001; **** = $P$ ≤ 0.0001). RNA-sequencing analyses were performed in bash with code generated in the lab and shown in the methods. The analysis and the statistics were performed via RStudio Version 2022.02.1 + 461.pro1 and the DESeq2 package v 4.3.1. The statistical significance was evaluated via padj method.

**Reporting summary**. Further information on research design is available in the Nature Portfolio Reporting Summary linked to this article.

## Data availability

The RNA-seq data for the *ceg1-AA* strains upon 45 min of treatment with DMSO or rapamycin are available in Gene Expression Omnibus (GEO) database with the accession number GSE213942. Source data for all graphs and charts in the main figures can be found in Supplementary Data 1 (Differentially expressed genes) and Supplementary Data 2 (qPCR values and microscopy quantifications). All other data are available from the corresponding author upon reasonable request.

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

## Acknowledgements

This work was funded by Sir Henry Dale Fellowship from the Wellcome Trust and the Royal Society (218537/Z/19/Z). O.Z. was supported by Midlands Doctoral Training Partnership by a studentship to O.Z. funded by BBSRC (grant number: BB/M01116X/1). D.H. is funded by EPSRC (grant EP/T002794/1). M.E. and M.A. are funded by BBSRC (BB/V015109/1). We thank Genomics Birmingham at the University of Birmingham for assistance with NGS.

## Author contributions

O.Z. performed most of the experiments (excluding approaches listed below), bioinformatics and data analyses. M.E. and M.A. performed microscopy, K.W. performed m7G cap immunoprecipitations and ribozyme analyses. D.H. supported bioinformatics. R.N.-M. supported data analysis. P.G. and O.Z. designed experimental approaches and wrote the manuscript.

## Competing interests

The authors declare no competing interests.
