## [Peer Review File · Communications Biology]

Reviewers' comments:

Reviewer #1 (Remarks to the Author):

Zanin et al.

In this manuscript, the authors investigate the effects on mRNA breakdown in *S. cerevisiae* of inactivating proteins required for capping. They use the "anchor-away" technique, which allows rapid depletion from the nucleus of the targeted protein.

They conclude that under conditions of inactivated mRNA capping, most of the uncapped mRNAs are rapidly degraded. However, a subgroup of mostly weakly expressed mRNAs are relatively resistant to decay and become enriched in the remaining mRNA population. The mRNA profile after some time of inactivated capping is similar to that of an *xrn1* mutant, indicating that Xrn1-dependent mRNA degradation is 5'-cap dependent.

The manuscript deals with a hitherto poorly investigated but fundamental question in RNA biology. Overall, there are some disagreements with earlier publications in the field. Based on results from thermosensitive *ceg1* mutants (lacking mRNA cap at the non-permissive temperature) and *ceg1 rat1* mutants (also lacking the exonuclease supposedly degrading non-capped mRNAs), the authors find evidence against a model where Rat1 degrades non-capped mRNAs co-transcriptionally. On the other hand, the only positive findings in this paper are the negative correlation between mRNA abundance and degradation rate, and the occurrence of a sequence motif in the non-capped mRNAs relatively resistant to degradation. This leaves a wide field of possibilities open, and no definitive model is proposed.

Generally, the experimental design is well suited to address this issue, and the results of general interest. It should however be noted that because of the nature of the issue at hand and available experimental techniques, the conclusions drawn are mostly indirect and by inference.

There are certain issues in the manuscript that need to be addressed before publication:

p. 10, middle: The relations between several parameters of the decaying mRNA population are investigated. It is said that (uncapped) mRNAs accumulating in the *ceg1* mutant are longer than average – on the other hand it is also said that this is in agreement with weakly expressed mRNAs being longer than average. If so, expression level is the determining factor, and length is just a confounding factor, unless the authors want to change their claim.

p. 10, last section: In the GO term analysis, were the enrichment for DNA-binding proteins corrected for the fact that the subset they were taken from already may have been enriched for that group?

An important take-home message is the strong enrichment of a sequence motif "[A/G]GAAA" in the population of mRNAs increasing in abundance. The authors simply state that the function of this sequence is unknown. Given the pronounced enrichment, it should be possible to perform indirect investigations to find out e.g. whether this motif has any resemblance to other identified motifs, if it co-occurs with other known patterns, if it is enriched in genes with some common function, etc. This could shed light on the significance of the findings in this paper.

The authors use one paper on normal RNA degradation rates in yeast to classify mRNAs into classes of stable, medium, and fast-degrading under normal conditions. This paper again uses the "anchor-away" method to remove RNA pol II from the nucleus. The half-life estimates from this method are far

higher than from more recent publications. The authors should correlate these half-life estimates with those derived from other methods to get a more reliable result.

A major concern about this work is the fact that by suddenly depleting the nucleus of its RNA capping capacity, the RNA degradation system will get flooded with substrates in the form of non-capped transcripts. This could lead to aberrant processes that would never occur under physiological circumstances. On p 24 near the end, they state: "...extensive degradation may saturate the exonucleases as previously shown for other aberrant RNAs...". In that case, where does that put the entire approach in this paper? Admittedly, it is hard to see how the question at hand could be studied without such disruptive techniques, but the authors should acknowledge and discuss this limitation more in depth.

Also in connection with this: Why would degradation of highly expressed mRNAs selectively attract degradation factors such that more weakly expressed mRNAs would be degraded more slowly? To make that claim, we would have to postulate a higher affinity of the degradation factors for highly expressed mRNAs – there is no evidence for that.

Minor remarks:

Abstract, next to last sentence: "...Rat1 is not involved... and consequently, the lack of capping does not affect the distribution of RNA polymerase II on the chromatin". The authors fail to mention here if this conclusion is based on a rat1 mutant or something else, and then the meaning of this sentence is lost.

In Fig. 2D, the decrease of abundance for mid-range expressed mRNAs in Ceg1-depleted cells for three individual genes is shown. This would be in accordance with the trend for that group as a whole, which was already shown earlier in the paper, but what new information does it bring to show these particular genes, seemingly chosen at random?

Reviewer #2 (Remarks to the Author):

Zanin et al. showed that the nuclear-depletion of Ceg1 upregulated a subset of transcripts with relatively low expression. The differential expression in Ceg1-AA was restored by the deletion of the cytoplasmic exonuclease XRN1, and the nuclear exonuclease Rat1 did not affect the degradation of cap-deficient transcripts and its transcription termination. Although these findings are unexpected and potentially interesting, strong evidence and careful discussions are lacking for countering the previous literature that Rat1 plays a role in degradation of uncapped transcripts (PMID: 20802481, 20188675, etc.). Also, the authors provided only circumstantial evidence for the mechanism for upregulation of lowly-expressed genes based on the comparison of genome-wide data. From these reasons, I would say that another journal may be suitable for this work.

Major points

1. As the author mentioned, the conclusion that Rat1 is irrelevant to quality control of cap-deficient transcripts should be derived from the experimental setup. Validations of the ceg1-AA system are required.

(1) The extent of nuclear depletion should be assessed using a microscopy or western blotting with nuclear fractionation. I recommend doing this for not only ceg1 but also other AA strains because it is possible that a small effect on cbp20/80 would be derived from the inefficient nuclear depletion.

(2) Although the authors examined the efficiency of loss of capping for ADH1 by ceg1-AA, it may vary

depending on transcripts. Genome-wide survey or individual analysis for all mRNAs tested in this manuscript should be necessary because the difference of uncapping is possible to affect the destiny of mRNAs, i.e., less uncapped transcripts may increase its level.

(3) Is it possible that uncapped transcripts are exported to cytoplasm together with Ceg1-AA recruited on ribosomes? If so, it is understandable that the cytoplasmic exonuclease mainly degrades such aberrant transcripts in this system.

2. The finding that the expression level causes opposite effects by the nuclear depletion of Ceg1 is intriguing, but more direct results would be necessary for the conclusion. Reporter assays should be performed to examine what kind of determinants such as sequence of a gene body, promoter strength, destabilization sequence, and motif identified in Fig. 2g are required for up- or down-regulation of transcripts.

3. Recapitulation by ribozyme-containing reporters would be better to examine the effect of cap deficiency on the differential destinies of transcripts, not only for the FMP27 gene. Also, several promoters should be examined because downregulation observed in Fig.5 may be derived from the alteration of basal transcription level by ADH1 promoter.

Minor points

1. The reason why ref.20 was not recapitulated in Fig. 3a even for the GAL1 gene is necessary to be discussed more carefully. Previously, 34°C was used as a non-permissive temperature, so is it possible that 37°C was too high for this analysis? I am also concerned about the increase or decrease of transcript level upon temperature shift.

2. Because global reduction of transcripts may be caused by ceg1-AA, it is hard to judge whether RNA amounts were increased without spike-in. Adding synthetic spike-in would be better. Alternatively, the normalization by internal spike-in that is proven not to be affected by uncapping can be done.

3. The relative location of the motif observed in Fig. 2g is valuable (cap-proximal, UTR, etc.).

4. In supplementary figure 3, enriched GO terms contain highly redundant information. For better readability, the GO terms would be better to summarize based on similar functions.

Responses to Reviewers' comments:

Reviewer #1 (Remarks to the Author):

Zanin et al.

*In this manuscript, the authors investigate the effects on mRNA breakdown in *S. cerevisiae* of inactivating proteins required for capping. They use the “anchor-away” technique, which allows rapid depletion from the nucleus of the targeted protein.*

*They conclude that under conditions of inactivated mRNA capping, most of the uncapped mRNAs are rapidly degraded. However, a subgroup of mostly weakly expressed mRNAs are relatively resistant to decay and become enriched in the remaining mRNA population. The mRNA profile after some time of inactivated capping is similar to that of an *xrn1* mutant, indicating that *Xrn1*-dependent mRNA degradation is 5'-cap dependent.*

*The manuscript deals with a hitherto poorly investigated but fundamental question in RNA biology. Overall, there are some disagreements with earlier publications in the field. Based on results from thermosensitive *ceg1* mutants (lacking mRNA cap at the non-permissive temperature) and *ceg1 rat1* mutants (also lacking the exonuclease supposedly degrading non-capped mRNAs), the authors find evidence against a model where *Rat1* degrades non-capped mRNAs co-transcriptionally. On the other hand, the only positive findings in this paper are the negative correlation between mRNA abundance and degradation rate, and the occurrence of a sequence motif in the non-capped mRNAs relatively resistant to degradation. This leaves a wide field of possibilities open, and no definitive model is proposed.*

Generally, the experimental design is well suited to address this issue, and the results of general interest. It should however be noted that because of the nature of the issue at hand and available experimental techniques, the conclusions drawn are mostly indirect and by inference.

Authors: We appreciate the comments. We have improved the manuscript accordingly to clarify our outcomes and conclusions. We do not propose a definitive model as the research into this area has just been initiated. However, as the Reviewer noticed, there is the gap in the knowledge and almost no research has been done in this fundamental area for more than a decade now, and we feel that this work should be disseminated to allow others to fully understand RNA turnover in eukaryotic cells.

There are certain issues in the manuscript that need to be addressed before publication:

1. Reviewer 1: p. 10, middle: *The relations between several parameters of the decaying mRNA population are investigated. It is said that (uncapped) mRNAs accumulating in the *ceg1* mutant are longer than average – on the other hand it is also said that this is in agreement with weakly expressed mRNAs being longer than average. If so, expression level is the determining factor, and length is just a confounding factor, unless the authors want to change their claim.*

Authors: We agree with the reviewer. The correlation between the gene length and accumulation in *ceg1-AA* additionally validates the observation that the stability of non-capped transcript is generally inversely proportional to their expression. We have changed the text to clarify this statement.

2. Reviewer 1: p. 10, last section: *In the GO term analysis, where the enrichment for DNA-binding proteins corrected for the fact that the subset they were taken from already may have been enriched for that group?*

Authors: We reconsidered this part of the manuscript and agree with both Reviewers that the GO analysis may be confusing for the reader. Therefore, we have removed this analysis from the current version of the manuscript.

3. Reviewer 1: *An important take-home message is the strong enrichment of a sequence motif “[A/G]GAAA” in the population of mRNAs increasing in abundance. The authors simply state that the function of this sequence is unknown. Given the pronounced enrichment, it should be possible to perform indirect investigations to find out e.g. whether this motif has any resemblance to other identified motifs, if it co-occurs with other known patterns, if it is enriched in genes with some common function, etc. This could shed light on the significance of the findings in this paper.*

Authors: We stated in the manuscript that we do not know what the function is of the motif because we did not find similarities to known sequences playing roles in RNA biology. Our search revealed that the motif resembles the binding sequence of the human NFAT (nuclear factor of activated T-cells) factor participating in T-cells immune response (Holtz-Heppelmann et al., 1998, *J. Biol. Chem.*; PMID: 9468493). NFAT displays functional similarity with the transcription factor Crz1 in *S. cerevisiae* (Thewes 2014, *Eukaryot. Cell.*; PMID: 24681686) however, we have not linked its function to RNA degradation pathways yet.

The motif is not enriched in genes with specific functions as the GO analysis of increased genes in *ceg1-AA* did not return any cohesive results.

As for the “co-occurrence with other known patterns”, we tested the distribution of any other motifs stabilising/destabilising Pol II transcripts (e.g., Nrd1-binding sites) and we did not see such co-occurrence with the sequences we analysed (not shown in the manuscript).

We also tested the distribution of the motif across the increased genes, and we found even distribution over the open reading frames (Fig. 3c). More detailed analysis of the motif will be published elsewhere.

4. Reviewer 1: *The authors use one paper on normal RNA degradation rates in yeast to classify mRNAs into classes of stable, medium, and fast-degrading under normal conditions. This paper again uses the “anchor-away” method to remove RNA pol II from the nucleus. The half-life estimates from this method are far higher than from more recent publications. The authors should correlate these half-life estimates with those derived from other methods to get a more reliable result.*

Authors: As the main source for our analyses, we chose data generated by Kevin Struhl’s lab (Geisberg et al., 2014; *Cell*; PMID: 24529382). To our knowledge, there are not any more recent, similarly reliable and comprehensive resources estimating mRNA half-lives in yeast in normal conditions. During the revision we used a dataset generated earlier in 2002, (Wang et al., 2002, *PNAS*; PMID: 11972065) which analysed 936 mRNAs (compared to 1002 analysed by Geisberg et al.) and subjected it to our analyses (Supplementary Fig. 2b,c). We found a similar trend in our data using these half-lives values. The abundance of mRNAs with longer half-life values increased in *ceg1-AA* strain while for those with shorter half-lives decreased. Thus, the outcomes of our analyses have been confirmed.

Also, we would like to point out that in most of our analyses we calculate “expression strength” using our own data obtained in WT cells.

5. Reviewer 1: *A major concern about this work is the fact that by suddenly depleting the nucleus of its RNA capping capacity, the RNA degradation system will get flooded with substrates in the form of non-capped transcripts. This could lead to aberrant processes that would never occur under physiological circumstances. On p 24 near the end, they state: "...extensive degradation may saturate the exonucleases as previously shown for other aberrant RNAs...". In that case, where does that put the entire approach in this paper? Admittedly, it is hard to see how the question at hand could be studied without such disruptive techniques, but the authors should acknowledge and discuss this limitation more in depth.*

Authors: The acute methods to deplete protein (e.g., anchor away system, degron tags) differ from classic approaches by the duration of time (minutes/hours versus days) and therefore allow us to investigate the most direct effects of protein depletion limiting the long-term indirect or compensation artefacts. The loss of function of either essential or non-essential protein, achieved by siRNA treatment, deletion, mutations, modification etc., or fast depletion methods create conditions which never happen in a healthy cell and thus, inform us about the potential function of the absent/inactive protein. We have substantially rewritten the manuscript and we hope that our experimental strategy is now clearer.

Reviewer 1: *Also in connection with this: Why would degradation of highly expressed mRNAs selectively attract degradation factors such that more weakly expressed mRNAs would be degraded more slowly? To make that claim, we would have to postulate a higher affinity of the degradation factors for highly expressed mRNAs – there is no evidence for that.*

Authors: We have rewritten this part of the discussion and removed the sentence about saturation of the exonucleases. We also cited a paper from the Cramer lab showing that highly expressed mRNAs have higher affinity to 3'-5' degradation enzymes (Sohrabi-Jahromi et al., 2019, *eLife*; PMID: 31135339) and therefore may be more extensively degraded than weakly expressed ones when the 5'-3' degradation pathway is affected by depletion of capping enzymes or deletion of *XRN1*. In contrast, low expressed genes rely mostly on 5'-3' degradation (Sohrabi-Jahromi et al., 2019, *eLife*; PMID: 31135339).

Minor remarks:

1. Reviewer 1: *Abstract, next to last sentence: "...Rat1 is not involved... and consequently, the lack of capping does not affect the distribution of RNA polymerase II on the chromatin". The authors fail to mention here if this conclusion is based on a rat1 mutant or something else, and then the meaning of this sentence is lost.*

Authors: We have rewritten the sentence.

2. Reviewer 1: *In Fig. 2D, the decrease of abundance for mid-range expressed mRNAs in Ceg1-depleted cells for three individual genes is shown. This would be in accordance with the trend for that group as a whole, which was already shown earlier in the paper, but what new information does it bring to show these particular genes, seemingly chosen at random?*

Authors: We followed an established "unwritten rule" in the transcriptional studies where metadata are supported by examples of single genes from the datasets. This shows that the observed phenotype is not a result of aberrant genomic features/analysis, low or locally spiked reads, or other artefacts. The genes are usually picked to be the most representative and informative. We have made changes to the manuscript to clarify this.

Reviewer #2 (Remarks to the Author):

Reviewer 2: Zanin *et al.* showed that the nuclear-depletion of *Ceg1* upregulated a subset of transcripts with relatively low expression. The differential expression in *Ceg1-AA* was restored by the deletion of the cytoplasmic exonuclease *XRN1*, and the nuclear exonuclease *Rat1* did not affect the degradation of cap-deficient transcripts and its transcription termination. Although these findings are unexpected and potentially interesting, strong evidence and careful discussions are lacking for countering the previous literature that *Rat1* plays a role in degradation of uncapped transcripts (PMID: 20802481, 20188675, etc.). Also, the authors provided only circumstantial evidence for the mechanism for upregulation of lowly-expressed genes based on the comparison of genome-wide data. From these reasons, I would say that another journal may be suitable for this work.

Authors: The Reviewer discusses our data with two papers:

We discussed extensively both papers in our manuscript. We would like to point out that in both papers the observations are based on 2-3 highly expressed genes. There is very little data regarding *Rat1* direct role in the degradation of uncapped transcripts, moreover, none of them contradicts our observation regarding RNA levels in the *ceg1-AA* mutant.

Jiao *et al.* (PMID: 20802481) show that *Rai1* has an essential role in clearing mRNAs with aberrant 5'-end (non-methylated) caps. The authors did not study the degradation of non-capped RNAs. They employed *in vitro* systems to investigate decapping activity of *Rai1* and tested the stability of capped but not methylated three highly expressed mRNAs *ACT1*, *PGK1* and *CYH2 (RPL28)* in *abd1* and *abd1/rai1* mutants. They also tested the stability of mRNA in *rai1* mutant during glucose and amino acid starvation. Thus, this paper indicates the role of *Rai1* and *Rat1* (not directly, as *Rat1* mutants are not tested in this paper) in the degradation of capped but not methylated mRNAs, which is different to our study.

Jimeno-Gonzalez *et al.* (PMID: 20188675) paper is focused mainly on *Rat1*-dependent transcription termination. The authors employed chromatin immunoprecipitation (ChIP) followed by qPCR analysis to test Pol II occupancy over the *YLR454* gene expressed from the *GAL1* promoter (Fig. 1, 2, and 4). The authors also tested *YLR454 (FMP27)* and *GAL1* mRNA stability in *rat1-1* and slow Pol II mutants (Fig. 4). For the experiments shown in Fig. 3B, the authors employed *ceg1-63* mutant to test Pol II distribution (using ChIP) over *YLR454* gene. The mRNA levels of *GAL1::YLR454 (GAL1::FMP27)* and *GAL1* in *ceg1-63* and *ceg1-63/rat1-1* were shown only in the supplementary Fig. S3 by Northern Blot (n=1). In this experiment (shown below, Fig. R1) the authors stopped the transcription of *GAL1::YLR454* and *GAL1* by shifting cells to glucose-containing medium and then tested the stability of the transcripts. In both cases *Rat1* mutation initially slowed down the degradation of non-capped RNAs up to 5-10 minutes from the shift, however, did not rescue the phenotype. This was apparent especially for *GAL1::YLR454* as after 30 min *YLR454* mRNA decreased to the same level in *ceg1-63* and *ceg1-63/rat1-1* mutants.

We performed a genome-wide study of the “steady-state” non-capped transcriptome and we did not block transcription or modify gene expression in the major experiments shown in our manuscript which is different to the referred papers.

We have made changes to the manuscript to make these points clear.

Figure S3

Figure R1. Figure S3 from Jimeno-Gonzalez et al.:

“Figure S3: Rat1p-Dependent Degradation of *YLR454* and *GAL1* RNA in *ceg1-63* Cells. Related to Figure 3. A) Chase experiments of *GAL1::YLR454* and *GAL1* RNA harvested from *ceg1-63* and *ceg1-63/rat1-1* strains under conditions as in Figure legend 3A. Upper panel: Northern blotting analysis using specific probes for *YLR454*, *GAL1*, and 25S RNAs. Lower panel: *YLR454* and *GAL1* RNA signals quantified relative to 25S rRNA. RNA levels at the 0 time point were set to 1.”

Major points:

1. Reviewer 2: As the author mentioned, the conclusion that *Rat1* is irrelevant to quality control of *cap*-deficient transcripts should be derived from the experimental setup. Validations of the *ceg1-AA* system are required.

Authors: We validated the experimental system as requested, see the next points below.

Reviewer 2: (1) The extent of nuclear depletion should be assessed using a microscopy or western blotting with nuclear fractionation. I recommend doing this for not only *ceg1* but also other AA strains because it is possible that a small effect on *cbp20/80* would be derived from the inefficient nuclear depletion.

Authors: In the revised version of the manuscript, we included microscopy analysis of nuclear depletions (Fig. 1c). We shifted GFP-fused *ceg1-AA*, *cbp20-AA* and *cbp80-AA* strains for 45 minutes to a medium containing either DMSO or rapamycin, fixed the cells and tested the location of the AA-tagged proteins. Moreover, we quantified the signal originating from nuclei and cytoplasm in both conditions (Fig. 1d). This analysis shows that the proteins were efficiently removed from the nucleus upon Rap treatment and that the AA system works very well in our experimental setup.

Reviewer 2: (2) *Although the authors examined the efficiency of loss of capping for ADH1 by ceg1-AA, it may vary depending on transcripts. Genome-wide survey or individual analysis for all mRNAs tested in this manuscript should be necessary because the difference of uncapping is possible to affect the destiny of mRNAs, i.e., less uncapped transcripts may increase its level.*

Authors: We followed the Reviewer's suggestion and tested mRNAs investigated in this manuscript for the presence of the cap in *ceg1-AA* after 45 min on rapamycin. We performed IP with an anti-m7G antibody and estimated the levels of capped mRNAs relative to the "DMSO" (WT) control using qPCR. The yeast RNA was spiked (1:5) with human mRNA and IP efficiency was normalised to *hGAPDH* levels. The levels of non-capped mRNAs were similarly affected for most of the tested genes. For example, we observed a reduction of capped mRNA by 78% for *ADH1* (which decreased in non-selected RNA fraction in *ceg1-AA*) and by 81% for *PML39* (which increased in non-selected RNA fraction *ceg1-AA*) (Fig. 1e and Supplementary Fig. 1b). The lowest ratio of capped/non-capped was observed for *PYK2* indicating that the difference in capping efficiency was not the reason for differential accumulation of mRNAs.

Reviewer 2: (3) *Is it possible that uncapped transcripts are exported to cytoplasm together with Ceg1-AA recruited on ribosomes? If so, it is understandable that the cytoplasmic exonuclease mainly degrades such aberrant transcripts in this system.*

Authors: It is an interesting direction; however, we speculate that this is not the case. For such long post-transcriptional binding the interaction of the capping complex with RNA would have to be very stable. We are not aware of such data as the structure of the Ceg1-Cet1 complex indicates that the enzyme is delivered to nascent RNA via Pol II (Gu et al., 2010, *Structure*; PMID: 20159466). Moreover, as observed for temperature-sensitive (ts) mutants the trafficking of uncapped from nucleus to cytoplasm is not affected (Fresco and Buratowski, 1996, *RNA*; PMID: 8718687).

2. Reviewer 2: *The finding that the expression level causes opposite effects by the nuclear depletion of Ceg1 is intriguing, but more direct results would be necessary for the conclusion. Reporter assays should be performed to examine what kind of determinants such as sequence of a gene body, promoter strength, destabilization sequence, and motif identified in Fig. 2g are required for up- or down-regulation of transcripts.*

Authors: We agree with the Reviewer that this is necessary to understand the metabolism of non-capped mRNAs. Following Reviewer's advice we analysed the sequences of gene bodies and found the differences in GC content and codon bias between the sets (Fig. 3). GC content as well as codon bias have been reported to play very important roles in overall regulation of RNA fates from the localization, storage to the selection of degradation pathways (e.g., Sohrabi-Jahromi et al., 2019, *eLife*; PMID: 31135339, Courel et al., 2019, *eLife*; PMID: 31855182; Kudla et al., 2006; *PLOS Biol.*; PMID: 16700628; Zhao et al., 2021 *PNAS*; PMID: 33526697). We discuss these findings in the revised version of the manuscript.

We also tested the distribution of the motif across the increased genes, and we found an even distribution over the open reading frames (Fig. 3c). A more detailed and comprehensive analysis of this stability determinant and its function will be described in a separate paper uncovering mechanisms of this regulation.

We agree with the Reviewer that more complex analyses would allow to fully understand the regulation, but it would also pre-empt the existing data. Multiple parameters (promoters, gene

body sequence, the presence of motifs) and complex interdependencies define the stability of the uncapped transcript. Thus the reporter assays would have to take into account many factors to recreate native conditions. Thus, at this early stage of this discovery using reporter constructs would be superficial and may result in artefacts. Such analysis is very important, but it will require a large-scale in-depth study and since we are limited by the volume of this manuscript, this is out of the scope of this work.

3. Reviewer 2: *Recapitulation by ribozyme-containing reporters would be better to examine the effect of cap deficiency on the differential destinies of transcripts, not only for the FMP27 gene. Also, several promoters should be examined because downregulation observed in Fig.5 may be derived from the alteration of basal transcription level by ADH1 promoter.*

Authors: We carefully revised the data regarding the ribozyme assays and we have significantly rewritten and toned down this part of the manuscript. As shown in the revised version of the manuscript, swapping the promoter is not sufficient to achieve the same properties/levels of the generated transcript as for the mRNA naturally expressed from that promoter. Thus, this reporter is used as an example that the generation of the correct 5' end is required for the efficient degradation of lowly transcribed mRNAs.

Most of the genes which are highly expressed are essential or play important roles in the cell. Thus, the insertion of the ribozyme into their 5' UTR and as a result, their knockdown may not be viable or may significantly affect cellular metabolism and fitness. Thus, testing mRNAs of highly expressed genes, is not feasible in this particular experimental setup.

We improved the text and added analysis of ribozyme construct upon capping enzymes depletion which clarifies the data.

Minor points

1. Reviewer 2: *The reason why ref.20 was not recapitulated in Fig. 3a even for the GAL1 gene is necessary to be discussed more carefully. Previously, 34°C was used as a non-permissive temperature, so is it possible that 37°C was too high for this analysis? I am also concerned about the increase or decrease of transcript level upon temperature shift.*

Authors: We would like to clarify that it was not our intention to either validate or disprove the work from the Jensen lab focused on Rat1 role in transcription. However, we also tested the mRNA levels at 34°C as requested by the Reviewer. The effect was similar to that observed at 37°C although the magnitude of the change was milder, as expected. The mutation of Rat1 did not rescue the levels of *ADH1*, *GAL1::FMP27* and *GAL1* mRNAs (which decreased) nor *NEL025c* ncRNA (which increased) (Supplementary Fig. 4a). We would like to point out again, that Jimeno-Gonzalez et al. did not focus on changes in mRNA levels. In this paper, the Rat1 mutation only partially rescued the *FMP27* and *GAL1* levels (only two mRNA tested by Northern Blot) in *ceg1-63* strain (as discussed and shown above) in the early time points of the temperature shift, and had very little, if any impact on longer (>30min) inactivation of Ceg1. We have rewritten the manuscript to clarify our data vs Jensen's paper.

2. Reviewer 2: *Because global reduction of transcripts may be caused by *ceg1-AA*, it is hard to judge whether RNA amounts were increased without spike-in. Adding synthetic spike-in would be better. Alternatively, the normalization by internal spike-in that is proven not to be affected by uncapping can be done.*

Authors: In our experimental approach, we employed a standard and widely used method for the detection of differentially expressed genes employing a model using the negative binomial distribution (Love et al., 2014, *Genome Biol.*; PMID: 25516281; >53K citation up to date).

Spike-ins are usually used where additional selection is applied prior to library selection e.g., in metabolic RNA labelling, RNA IPs etc. We would like to also point out that in our data many RNAs did not change their levels or decreased. It is unlikely that in such a situation only the analysis of RNA accumulation would be faulty in our bioinformatics pipeline. Moreover, our qPCR analyses validated the RNA-seq outputs. Here, we normalised mRNA levels to rRNA which is very stable and not affected by lack of the capping processes. We also included the experiment where we spiked-in yeast RNA with human RNA and used GAPDH to normalise the efficiency of reverse transcription between the samples (Supplementary Fig. 1e). These results are the same as in for non spiked-in experiments, therefore, we feel that repeating the sequencing experiments with the spike-ins is not necessary and taking into account the costs of such analysis, not sustainable.

3. Reviewer 2: *The relative location of the motif observed in Fig. 2g is valuable (cap-proximal, UTR, etc.).*

Authors: We tested the distribution of the motif across the increased genes, and we found a rather even distribution over the open reading frames (Fig. 3c). More detailed analysis of the motif will be published elsewhere.

4. Reviewer 2: *In supplementary figure 3, enriched GO terms contain highly redundant information. For better readability, the GO terms would be better to summarize based on similar functions.*

Authors: We reconsidered this part of the manuscript and agree with both Reviewers that the GO analysis may be confusing for the reader and therefore we have removed this analysis from the current version of the manuscript.

Reviewers' comments:

Reviewer #1 (Remarks to the Author):

In their revised manuscripts, the authors have adequately addressed all my concerns, except for some minor issues, as listed below.

After those items have been reconsidered, I would find the paper acceptable for publication.

Reviewer #1 (Remarks to the Author):

There are certain issues in the manuscript that need to be addressed before publication:

1. Reviewer 1: p. 10, middle: The relations between several parameters of the decaying mRNA population are investigated. It is said that (uncapped) mRNAs accumulating in the *ceg1* mutant are longer than average – on the other hand it is also said that this is in agreement with weakly expressed mRNAs being longer than average. If so, expression level is the determining factor, and length is just a confounding factor, unless the authors want to change their claim.

Authors: We agree with the reviewer. The correlation between the gene length and accumulation in *ceg1-AA* additionally validates the observation that the stability of non-capped transcript is generally inversely proportional to their expression. We have changed the text to clarify this statement.

- The revised text now emphasizes the role of expression level, which is adequate.

2. Reviewer 1: p. 10, last section: In the GO term analysis, where the enrichment for DNA-binding proteins corrected for the fact that the subset they were taken from already may have been enriched for that group?

Authors: We reconsidered this part of the manuscript and agree with both Reviewers that the GO analysis may be confusing for the reader. Therefore, we have removed this analysis from the current version of the manuscript.

- OK.

3. Reviewer 1: An important take-home message is the strong enrichment of a sequence motif "[A/G]GAAA" in the population of mRNAs increasing in abundance. The authors simply state that the function of this sequence is unknown. Given the pronounced enrichment, it should be possible to perform indirect investigations to find out e.g. whether this motif has any resemblance to other identified motifs, if it co-occurs with other known patterns, if it is enriched in genes with some common function, etc. This could shed light on the significance of the findings in this paper.

Authors: We stated in the manuscript that we do not know what the function is of the motif because we did not find similarities to known sequences playing roles in RNA biology. Our search revealed that the motif resembles the binding sequence of the human NFAT (nuclear factor of activated T-cells) factor participating in T-cells immune response (Holtz-Heppelmann et al., 1998, J. Biol. Chem.; PMID: 9468493). NFAT displays functional similarity with the transcription factor Crz1 in *S. cerevisiae* (Thewes 2014, Eukaryot. Cell.; PMID: 24681686) however, we have not linked its function to RNA degradation pathways yet.

The motif is not enriched in genes with specific functions as the GO analysis of increased genes in *ceg1-AA* did not return any cohesive results.

As for the "co-occurrence with other known patterns", we tested the distribution of any other motifs

stabilising/destabilising Pol II transcripts (e.g., Nrd1-binding sites) and we did not see such co-occurrence with the sequences we analysed (not shown in the manuscript).

We also tested the distribution of the motif across the increased genes, and we found even distribution over the open reading frames (Fig. 3c). More detailed analysis of the motif will be published elsewhere.

- Even if very little is understood about this motif, it would be worthwhile including this information here (similarity to NFAT).

4. Reviewer 1: The authors use one paper on normal RNA degradation rates in yeast to classify mRNAs into classes of stable, medium, and fast-degrading under normal conditions. This paper again uses the "anchor-away" method to remove RNA pol II from the nucleus. The half-life estimates from this method are far higher than from more recent publications. The authors should correlate these half-life estimates with those derived from other methods to get a more reliable result.

Authors: As the main source for our analyses, we chose data generated by Kevin Struhl's lab (Geisberg et al., 2014; Cell; PMID: 24529382). To our knowledge, there are not any more recent, similarly reliable and comprehensive resources estimating mRNA half-lives in yeast in normal conditions. During the revision we used a dataset generated earlier in 2002, (Wang et al., 2002, PNAS; PMID: 11972065) which analysed 936 mRNAs (compared to 1002 analysed by Geisberg et al.) and subjected it to our analyses (Supplementary Fig. 2b,c). We found a similar trend in our data using these half-lives values. The abundance of mRNAs with longer half-life values increased in *ceg1-AA* strain while for those with shorter half-lives decreased. Thus, the outcomes of our analyses have been confirmed.

Also, we would like to point out that in most of our analyses we calculate "expression strength" using our own data obtained in WT cells.

- The authors could check e.g. Chan et al. eLife 7: e32536 (2018) for a more modern estimate of RNA half-life to see if that would affect the results. "Expression strength" does not give information about RNA decay rates.

5. Reviewer 1: A major concern about this work is the fact that by suddenly depleting the nucleus of its RNA capping capacity, the RNA degradation system will get flooded with substrates in the form of non-capped transcripts. This could lead to aberrant processes that would never occur under physiological circumstances. On p 24 near the end, they state: "...extensive degradation may saturate the exonucleases as previously shown for other aberrant RNAs...". In that case, where does that put the entire approach in this paper? Admittedly, it is hard to see how the question at hand could be studied without such disruptive techniques, but the authors should acknowledge and discuss this limitation more in depth.

Authors: The acute methods to deplete protein (e.g., anchor away system, degron tags) differ from classic approaches by the duration of time (minutes/hours versus days) and therefore allow us to investigate the most direct effects of protein depletion limiting the long-term indirect or compensation artefacts. The loss of function of either essential or non-essential protein, achieved by siRNA treatment, deletion, mutations, modification etc., or fast depletion methods create conditions which never happen in a healthy cell and thus, inform us about the potential function of the absent/inactive protein. We have substantially rewritten the manuscript and we hope that our experimental strategy is now clearer.

- The revised version is indeed much improved in this respect.

Reviewer 1: Also in connection with this: Why would degradation of highly expressed mRNAs selectively attract degradation factors such that more weakly expressed mRNAs would be degraded more slowly? To make that claim, we would have to postulate a higher affinity of the degradation factors for highly expressed mRNAs – there is no evidence for that.

Authors: We have rewritten this part of the discussion and removed the sentence about saturation of the exonucleases. We also cited a paper from the Cramer lab showing that highly expressed mRNAs have higher affinity to 3'-5' degradation enzymes (Sohrabi-Jahromi et al., 2019, eLife; PMID: 31135339) and therefore may be more extensively degraded than weakly expressed ones when the 5'-3' degradation pathway is affected by depletion of capping enzymes or deletion of XRN1. In contrast, low expressed genes rely mostly on 5'-3' degradation (Sohrabi-Jahromi et al., 2019, eLife; PMID: 31135339).

- OK.

Minor remarks:

1. Reviewer 1: Abstract, next to last sentence: "...Rat1 is not involved... and consequently, the lack of capping does not affect the distribution of RNA polymerase II on the chromatin". The authors fail to mention here if this conclusion is based on a rat1 mutant or something else, and then the meaning of this sentence is lost.

Authors: We have rewritten the sentence.

- OK.

2. Reviewer 1: In Fig. 2D, the decrease of abundance for mid-range expressed mRNAs in Ceg1-depleted cells for three individual genes is shown. This would be in accordance with the trend for that group as a whole, which was already shown earlier in the paper, but what new information does it bring to show these particular genes, seemingly chosen at random?

- Authors: We followed an established "unwritten rule" in the transcriptional studies where metadata are supported by examples of single genes from the datasets. This shows that the observed phenotype is not a result of aberrant genomic features/analysis, low or locally spiked reads, or other artefacts. The genes are usually picked to be the most representative and informative. We have made changes to the manuscript to clarify this.

- This is acceptable.

Reviewer #2 (Remarks to the Author):

This reviewer appreciated the authors' efforts in improving this manuscript. Almost all my concerns have been addressed by adding the validation data. Also, I apologize for not understanding the previous literature well. The additional explanation and discussion are helpful for readers to understand the difference from the earlier studies. Therefore, I recommend publishing this manuscript.

It would be better if other experiments such as reporter assays could validate the trend in RNA change in the *ceg1* mutant, but I agree with the authors' opinion that various factors can affect the results and it may take time. I look forward to future research.

Responses to Reviewers' comments:

Below we listed two minor issues raised by the Reviewer 1:

Reviewer #1 (Remarks to the Author):

In their revised manuscripts, the authors have adequately addressed all my concerns, except for some minor issues, as listed below.

After those items have been reconsidered, I would find the paper acceptable for publication.

1. Reviewer 1: *Even if very little is understood about this motif, it would be worthwhile including this information here (similarity to NFAT).*

Authors: We included this information into the manuscript (page 10).

2. Reviewer 1: *The authors could check e.g. Chan et al. eLife 7: e32536 (2018) for a more modern estimate of RNA half-life to see if that would affect the results.*

“Expression strength” does not give information about RNA decay rates.

Authors: We appreciate this comment as further investigation of yeast mRNA half-lives resulted in unexpected outcomes. In our analyses we used data from Geisberg et al., 2014; *Cell*; PMID: 24529382; (254 citations currently) and Wang et al., 2002, *PNAS*; PMID: 11972065 (874 citations) which used Pol II mutants to arrest transcription followed by estimation of mRNA half-lives. In the study by Chan et al. 2018; *eLife*; PMID: 30192227 (186 citations) the authors used *in-vivo* metabolic RNA labelling to assess the degradation rates.

The newer study does not comment on the previous findings therefore, we compared the datasets from Geisberg et al., 2014 with the data generated by Chan et al. 2018 (Fig. R1). We found that the times of mRNA half-lives profoundly differ between the studies and often contradict each other. The plot in Fig. R1a shows the correlation between the two datasets. The R^2 value clearly indicates the lack of similarity and even implies an inverse correlation. We plotted individual examples (Fig. R1b) showing discrepancies in half-lives for the genes we used in our work. Although the half-life for highly expressed gene *ADH1* is similar, the half-lives of lowly expressed genes like *PML39* or *STP2* differ from 20 to 65 times between the datasets. This entirely changes the stability classification of low abundant transcripts relative to highly expressed genes in both studies. These dissimilarities seem to be introduced by the experimental approaches used. For example, half-life times distribution by Miller et al., 2011; *Mol Syst Biol*; where *in-vivo* metabolic RNA labelling was also used, correlate well with Chan et al. 2018 data although the half-life times values are significantly different in length (Fig. R1c). Similarly, data by Geisberg et al., 2014 correlates better with Wang et al., 2002 (Fig. R1d).

We appreciate that *in-vivo* metabolic RNA labelling (which we use in my lab to analyse nascent transcriptomes) allows to analyse mRNAs in “non-invasive” and native conditions however, the purification of low abundant labelled mRNAs may be sometimes challenging. Overall, in our opinion, it is not clear which approach is correct and since the half-lives of non-capped mRNAs is only one of the many features

describing the affected genes in our analyses, we decided to remove this data from the manuscript. Our major determinant is “expression strength” which is the level of the steady-state mRNAs, based on the mRNA accumulation in isogenic WT cells. Thus, the lack of information on their half-lives does not change the outcomes of our discoveries. We made corrections in the manuscript to point out that estimation of half-life times for increased or decreased non-capped mRNA is not feasible since it is not clear what are the true mRNA half-lives values in yeast as they significantly vary between published studies.

We would like to thank the Reviewer for bringing this to our attention as this significantly helped to correct the manuscript.

Figure 1R. **a** Correlation between two data sets using different methods to calculate mRNAs half-lives (Chan *et al.*, 2018 and Geisberg *et al.*, 2014). Negative R^2 indicates inverse correlation. **b** The ratio of half-life values from Chan *et al.*, 2018 and Geisberg *et al.*, 2014 for selected genes used in our study. **c,d** Correlation between two mRNAs half-lives data sets Chan *et al.*, 2018 and Miller *et al.*, 2011(c) and Wang *et al.*, 2002 and Geisberg *et al.*, 2014 (d).